# Characteristics of Greenhouse Gas Concentrations Derived from Ground-based FTS Spectra at Anmyeondo, Korea

Young-Suk Oh[1,2*], Samuel Takele Kenea[1], Tae-Young Goo[1], Kyu-Sun Chung[2], Jae-Sang Rhee[1], Mi-Lim Ou[6], Young-Hwa Byun[1], Paul O. Wennberg[4,5], Matthäus Kiel[4], Joshua P. DiGangi[7], Glenn S. Diskin[7], Voltaire A. Velazco[3] and David W. T. Griffith[3]

1. Climate Research Division, National Institute of Meteorological Sciences (NIMS), Jeju-do, Republic of Korea
2. Department of Electrical Eng. & Centre for Edge Plasma Science, Hanyang University, Seoul, Republic of Korea
3. School of Chemistry, University of Wollongong, Northfields Ave, Wollongong, NSW 2522, Australia
4. Division of Geological and Planetary Sciences, California Institute of Technology, 1200E. California Boulevard, Pasadena, CA, 91125, USA
5. Division of Engineering and Applied Sciences, California Institute of Technology, 1200E. California Boulevard, Pasadena, CA, 91125, USA
6. Climate Change Monitoring Division, Korea Meteorological Administration, Seoul, Republic of Korea
7. NASA Langley Research Center Building 1250, Mail Stop 401A Hampton, VA 23681 United States

*Correspondence to: Young-Suk Oh (ysoh306@gmail.com )

**Abstract.**

Since the late 1990s, the meteorological observatory established in Anmyeondo (36.5382° N, 126.3311° E, and 30 m above mean sea level), has been monitoring several greenhouse gases such as $CO_2$, $CH_4$, $N_2O$, CFCs, and $SF_6$, as a part of the Global Atmosphere Watch (GAW) Program. A high resolution ground-based (g-b) Fourier Transform Spectrometer (FTS) was installed at this observation site in 2013, and has been operated within the frame work of the Total Carbon Column Observing Network (TCCON) since August, 2014. The solar spectra recorded by the g-b FTS cover the spectral range 3,800 to 16,000 $cm^{-1}$ at a resolution of 0.02 $cm^{-1}$. In this work, the GGG2014 version of the TCCON standard retrieval algorithm was used to retrieve total column average $CO_2$ and $CH_4$ dry mole fractions ($XCO_2$, $XCH_4$) and from the FTS spectra. Spectral bands of $CO_2$ (at 6220.0 and 6339.5 $cm^{-1}$ centre wavenumbers, $CH_4$ at 6002 $cm^{-1}$ wavenumber, and $O_2$ near 7880 $cm^{-1}$ ) were used to derive the $XCO_2$ and $XCH_4$. In this paper, we provide comparisons of $XCO_2$ and $XCH_4$ between the aircraft observations and g-b FTS over Anmyeondo station. A comparison of 13 coincident observations of $XCO_2$ between g-b FTS and OCO-2 (Orbiting Carbon Observatory) satellite measurements are also presented for the measurement period between February 2014 and November 2017. OCO-2 observations are highly correlated with the g-b FTS measurements ($r^2$=0.884) and exhibited a small positive bias (0.189 ppm). Both data set capture seasonal variations of the target species with maximum and minimum values in spring and late summer, respectively. In the future, it is planned to further utilize the FTS measurements for the evaluation of satellite observations such as Greenhouse Gases Observing Satellite (GOSAT, GOSAT-2). This is the first report of the g-b FTS observations of $XCO_2$ species over the Anmyeondo station.

**Key words:** Aircraft, $XCO_2$, OCO-2, TCCON, Infrared spectra

## 1. Introduction

Monitoring of greenhouse gases (GHGs) is a crucial issue in the context of global climate change. Carbon dioxide ($CO_2$) is one of the key greenhouse gases and its global annual mean concentration has increased rapidly from 278 to 400 ppm since the preindustrial data of 1750 (WMO greenhouse gas bulletin, 2016). Radiative forcing due to changes in atmospheric $CO_2$ accounts for approximately 65 % of the total change in radiative forcing by long-lived GHGs (Ohyama et al., 2015 and reference therein). Human activities such as burning of fossil fuels and land use change are the primary drivers of the continuing increase in atmospheric greenhouse gases and the gases involved in their chemical production (Kiel et al., 2016 and reference therein). There is a global demand for accurate and precise long-term measurements of greenhouse gases.

In the field of remote sensing techniques, solar absorption infrared spectroscopy has been increasingly used to determine changes in atmospheric constituents. Today, a number of instruments deployed on various platforms (ground-based and space-borne) have been operated for measuring GHGs such as $CO_2$. The g-b FTS at the Anmyeondo station has been measuring several atmospheric GHG and other gases such as $CO_2$, $CH_4$, CO, $N_2O$, and $H_2O$ operated within the framework of the Total Carbon Column Observing Network (TCCON). $XCO_2$ retrievals from the g-b FTS have been reported at different TCCON sites (e.g, Ohyama et al., 2009; Deutscher et al., 2010; Messerschmidt et al., 2010, 2012; Miao et al., 2013; Kivi and Heikkinen, 2016, Velazco et al. 2017). TCCON achieves accuracy and precision in measuring the column averaged dry air mole fraction of $CO_2$ ($XCO_2$), of about 0.25 %, or better than 1 ppm (Wunch et al., 2010), which is essential to retrieve information about sinks and sources, as well as validating satellite products (Rayner and O'Brien, 2001; Miller et al., 2007). Precision for $XCO_2$ of 0.1 % can be achieved during clear sky conditions (Messerschmidt et al., 2010; Deutscher et al., 2010). The network aims to improve global carbon cycle studies and supply the primary validation data of different atmospheric trace gases for space-based instruments, e.g., the Orbiting Carbon Observatory 2 (OCO-2), the Greenhouse Gases Observing Satellite (GOSAT, GOSAT-2) (Morino et al., 2011; Frankenberg et al., 2015).

This study is focused on the initial characteristics of $XCO_2$ retrievals from g-b FTS spectra over the Anmyeondo station, and comparison with in situ aircraft overflights and the OCO-2 satellite. The FTS spectra have been processed using the TCCON standard GGG2014 (Wunch et al., 2015) retrieval software. One of the unique aspects in this work is a new homemade addition to our g-b FTS instrument that reduces the solar intensity variations from the 5% maximum allowed in TCCON to less than 2%. This paper presents an introduction to the instrumentation and measurement site, and provides initial results and discussion followed by conclusions.

## 2 Station and instrumentation

### 2.1 Station description

The g-b FTS observatory was established in 2013 at the Anmyeondo (AMY) station, located at 36.32˚ N, 126.19˚ E, and 30 m above sea level. This station is situated on the west coast of the Korean Peninsula, 180 km SE of Seoul, the capital city of Republic of Korea. Figure 1 displays the Anmyeondo station. It is also a regional GAW (Global Atmosphere Watch) station that is operated by the Climate Change Monitoring Network of KMA (Korean Meteorological Administration). The AMY station has

been monitoring various atmospheric parameters such as greenhouse gases, aerosols, ultraviolet radiation, ozone, and precipitation since 1999. The total area of the Anmyeondo island is estimated to be ~88 km$^2$ and approximately 1.25 million people reside on the island. Some of the residents over this area are engaged in agricultural activities. Vegetated areas consisting of mainly pine trees are located in and around the FTS observatory. The topographic feature of the area is one of low level hills, on

average about 100 m above sea level. The minimum temperature in winter season is on average 2.7 ˚C, and the maximum temperature is about 25.6 ˚C during summer. Average annual precipitation amount is 1,155 mm; with snow in winter. The site has been formally designated as a provisional TCCON site since August 2014. Full acceptance requires calibration via overflights with WMO-calibrated in situ vertical profiles, as described in this paper. The AMY Station's TCCON wiki page can be found at:

[https://tccon-wiki.Anmyeondo.edu]

### 2.2 G-b FTS instrument

Solar spectra are acquired using a Bruker IFS 125HR spectrometer (Bruker Optics, Germany) under the guidelines set by TCCON. Currently, our g-b FTS instrument operation is semi-automated for taking

the routine measurements under clear sky conditions. It is planned to make an FTS operation mode fully automated in 2018. The solar tracker (A547, Bruker Optics, Germany) is mounted inside a remotely controlled protective dome. The tracking ranges in azimuthal and elevation angles are about 0 to 315 and -10 to 85 degrees, respectively, while the tracking speed is about 2 degrees per second. The tracking accuracy of ±4 minutes of arc is achieved by the Camtracker mode which centres an image of the sun

onto the spectrometer's input field stop. Under clear sky conditions, the dome is opened and set to an automatic tracking mode, in which the mirrors are initially moved to the calculated solar position, then. The camtracker control is activated in such a way that the mirrors are finely and continuously controlled to fix the beam onto the entrance stop of the spectrometer. Figure 2 displays an overview of the general data acquisition system. This ensures that all spectra are recorded under clear weather conditions.

The spectrometer is equipped with two room temperature detectors; an Indium-Gallium-Arsenide (InGaAs) detector, which covers the spectral region from 3,800 to 12,800 cm$^{-1}$, and Silicon (Si) diode detector (9,000 – 25,000 cm$^{-1}$) used in a dual-acquisition mode with a dichroic optic (Omega Optical,

10,000 cm$^{-1}$ cut-on). A red longpass filter (Oriel Instruments 59523; 15,500 cm$^{-1}$ cut-on) prior to the Si diode detector blocks visible light, which would otherwise be aliased into the near-infrared spectral domain. TCCON measurements are routinely recorded at a maximum optical path difference (OPD$_{max}$) of 45 cm leading to a spectral resolution of 0.02 cm$^{-1}$ (0.9/max OPD). Two scans, one forward and one backward, are performed and individual forward-backward interferograms are recorded. As an example, Figure 3 shows a single spectrum recorded on 4 October 2014 with a resolution of 0.02 cm$^{-1}$. A single forward-backward scan in one measurement takes about 112 s. Measurement setting for the Anmyeondo g-b FTS spectrometer of the Bruker 125HR model is summarized in Table 1. The pressure inside the FTS is kept at 0.1 to 0.2 hPa with an oil-free vacuum pump to maintain the stability of the system and to ensure clean and dry conditions.

## 2.3 Characterization of FTS-instrumental line shapes

For the accurate retrieval of total column amounts of the species of interest, a good alignment of the g-b FTS is essential. The instrument line shape (ILS) retrieved from the regular HCl cell measurements is an important indicator of the status of the FTS's alignment (Hase et al., 1999). The analyses of the measurements were performed using a spectrum fitting algorithm (LINEFIT14 software) (Hase et al., 2013). In Figure 4 we show time series of the modulation efficiency (right panel) and phase error (rad) (left panel) from the HCl cell measurement in the period of October 2013 to September 2017 using a tungsten lamp as light source. Modulation amplitudes for TCCON-acceptable alignment should be within 5 % of the ideal case (100%) at the maximum optical path difference (Wunch et al., 2011). In our g-b FTS measurements, it is found that the maximum loss of modulation efficiency is within 1 %, close to the ideal value. The phase errors are less than ± 0.0001 rad. Hase et al. (2013) reported that this level of small disturbances from the ideal value of the modulation efficiency is common to all well-aligned instruments. This result confirmed that the g-b FTS instrument is well aligned and has remained stable during the whole operation period.

We also confirmed that the ILS was not affected by the variable aperture (OASIS) during the operation of this system (see section 2.5). The modulation efficiency and phase error were estimated to be 99.98 % and 0.0001 rad. Sun et al. (2017) reported the detailed characteristics of the ILS with respect to applications of different optical attenuators to FTIR spectrometers within the TCCON and NDACC networks. They used both lamp and sun as light sources for the cell measurements, which were conducted after the insertion of five different attenuators in front of and behind the interferometer.

## 2.4 Data processing

Using the TCCON standard retrieval strategy, we have derived the column-averaged dry-air mole fractions $CO_2$ (X$CO_2$) and other atmospheric gases ($O_2$, CO, $CH_4$, $N_2O$, and $H_2O$) using the GFIT

algorithm and software. The spectral windows used for the retrieval of $CO_2$ and $O_2$ are given in Table 2. The TCCON standard GGG2014 (version 4.8.6) retrieval software was used to obtain the abundance of the species from FTS spectra (Wunch et al., 2015). $XCO_2$ is derived from the ratio of retrieved $CO_2$ column to retrieved $O_2$ column,

$$XCO_2 = \frac{CO_{2\ column}}{O_{2\ column}} \times 0.2095 \,, \tag{1}$$

Computing the ratio using Eq. (1) minimizes systematic and correlated errors such as errors in solar zenith angle pointing error, surface pressure, and instrumental line shape that may exist in the retrieved $CO_2$ and $O_2$ columns (Washenfelder et al., 2006, Messerschmidt et al., 2012). The top panel of Figure 5 depicts the time series of laser sampling error (LSE) obtained from InGaAs spectra at the Anmyeondo FTS station in the measurement period of February 2014 to December 2016. LSE is due to inaccuracies in the laser sample timing, which have been reduced to acceptable levels by the instrument manufacturer. In the AMY FTS, the LSE is small and centered around zero. Slightly large LSE values were shown on 10 March, 2014 (see top panel of Fig. 5). On this date, we conducted the laser adjustment in FTS.

$X_{air}$ is the ratio of atmospheric pressure to total column $O_2$, scaled such that for a perfect measurement $X_{air} = 1.0$. $X_{air}$ is a useful indicator of the quality of measurements and the instrument performance (Wunch et al., 2015). Due to spectroscopic limitations there is a TCCON wide bias ($X_{air} \sim 0.98$) and small solar zenith angle (SZA) dependence. The retrieval of $X_{air}$ deviating more than 1% from the TCCON-wide mean value of 0.98 would suggest a systematic error. The time series of $X_{air}$ is shown in the bottom panel of Figure 5. The $X_{air}$ record reveals that the instrument has been stable during the measurement period. It shows that the values of $X_{air}$ fluctuate between 0.974 and 0.985, and the mean value is 0.982 with a standard deviation of 0.0015 in which the scatter for $X_{air}$ is about 0.15 %. The low variability in time series of $X_{air}$ indicates the stability of the measurements.

## 2.5 Operational Automatic System for the Intensity of Sunray (OASIS) effect on the retrieval results

The OASIS system was developed for improving the quality of the spectra recorded by the spectrometer by maintaining a constant signal level. OASIS is beneficial for minimizing the variability that may be induced in the spectra due to intensity fluctuations of the incoming solar radiation that reaches the instrument. The main function of the OASIS is to control an aperture diameter in the parallel-inlet beam to the interferometer. This aperture is placed inside the OASIS system, in the parallel input solar beam external to the FTS. The fundamental purpose of this system is to optimize the measurement of solar spectra by reducing the effect of the fluctuations of the intensity of the incoming light due to changes in thin clouds along the line of sight over the measurement site. The maximum threshold value of the solar

intensity variation (SIV) is 5 %, the TCCON standard value (Ohyama et al., 2015). This value has been reduced to ≤ 2 % in our case by introducing the OASIS system to our g-b FTS since December 2014.

In order to assess the impact of the OASIS system on the retrieval results of $XCO_2$ and $XCH_4$, we have conducted experiments on recording alternate FTS spectra with and without operation of this system under clear sky conditions. As an example, Figure 6 depicts the retrieval results of $XCO_2$ (left panel) and $XCH_4$ (right panel) as a function of time (KST, UTC+9), taken November 23, 2017 with OASIS on (blue) and off (red) positions. Mean differences of 0.12 ppm for $XCO_2$ and 7.0 x $10^{-4}$ ppm for $XCH_4$ were found between OASIS on and off position (i.e., with and without operating of OASIS system). This suggests that the impact of OASIS system on the retrieval is negligible.

## 2.6 Aircraft observation campaigns over Anmyeondo station

### 2.6.1 Aircraft instrumentation

In this section, we present a comparison between aircraft in-situ observations and g-b FTS column measurements over the Anmyeondo station. In situ profiles were conducted over Anmyeondo station by the National Institute of Meteorological Sciences (King Air C90) and as part of the KORUS-AQ campaign from NASA's DC8 (https://www-air.larc.nasa.gov/missinns/korus-aq). For the NIMS profiles, the flight take-off and landing was carried out from Hanseo University which is approximately 5 km away from the Anmyeondo FTS station. The aircraft was equipped with a Wavelength Scanned Cavity Ring Down Spectrometer (CRDS; Picarro, G2401-m), (see Fig. 7) providing mixing ratio data recorded at 0.3 Hz intervals. The position of the aircraft was monitored by GPS, and information on the outside temperature, static pressure, and ground speed was provided by instruments carried on the plane. The temperature and pressure of the gas sample have to be tightly controlled at 45 ℃ and 140 Torr in the CRDS, which leads to highly stable spectroscopic features (Chen et al., 2010). Any deviations from these values cause a reduction of the instrument's precision. Data recorded beyond the range of variations in cavity pressure and temperature were discarded in this analysis. Variance of the cavity pressure and temperature during flight results in variance in the $CO_2$ and $CH_4$ mixing ratios. The Picarro CRDS instrument has been regularly calibrated with respect to the standard gases within the error range recommended by the World Meteorological Organization. Measurements were made in wet air, and dry air mixing ratios were derived following the method described in Chen et al. (2010). Water was measured and its effect was accounted for in the column integration of $CO_2$ and $CH_4$

On NASA's DC8, $CO_2$ was measured by the Atmospheric Vertical Observations of $CO_2$ in the Earth's troposphere (AVOCET) instrument, a non-dispersive IR spectrometer (Vay et al., 2009) with an uncertainty of 0.25 ppm, $CH_4$ was measured by the Differential Absorption of CO Measurement (DACOM) instrument, a mid-IR absorption sensor (Sachse at al., 1987) with an accuracy of 1% and a

precision of 1 ppb. Both instruments were calibrated in-flight with standard gases traceable to the respective World Meteorological scales. The aircraft static pressure and altitude were recorded via a pressure transducer and radar altimeter, respectively, recorded by the aircraft data system. As with the NIMS profiles, the vertical profiles of $CO_2$ and $CH_4$ mixing ratio were obtained during a downward
flight centred on the Anmyeondo.

### 2.6.2 Aircraft $CO_2$ and $CH_4$ data

The NIMS vertical profiles of $CO_2$ and $CH_4$ mixing ratio were obtained during a downward spiral flight centred over the Anmyeondo FTS station, on 29 October and 12 November, 2017. As   an example, the
225 flight trajectory is shown in the left panel of Figure 8 while the profiles of $CO_2$ and $CH_4$ from flight during the ascent and descent on 29 October, 2017 are depicted in the middle and right panels of  Figure 8, respectively. All flights were performed under clear sky conditions. The campaign was p-erformed for 2 hours on both days. Specifically, the respective measurements were taken from 11:00:37 to 12:03:25 K ST (UTC+9) and from 13:58:58 to 15:19:40 KST on 29 October, 2017 and similarly from 11:12:20 to 1
2:13:00 KST and from 14:14:46 to 15:14:46 KST on 12 November, 2017. The altitude range of the aircr aft measurements was limited to approximately 0.1 to a 9.1 km. We constructed the complete $CO_2$ and $CH_4$ profiles in a similar way as performed by Deutscher et al. 2010; Miyamoto et al. (2013); Ohyama e t al. (2015).

For both $CO_2$ and $CH_4$ profiles, we have used in-situ surface data (AMY GAW station) to complement
the aircraft profiles close to surface level, and above the aircraft ceiling, the mole fractions throughout t he altitude range between the uppermost aircraft and the tropopause is assumed to be the same as at the highest aircraft measurement level because of lack of data. This extrapolation produces the largest uncertainty in the in situ column estimate.  For this analysis, the tropopause height was derived from N OAA National Centers for Environmental Prediction/National Center for Atmospheric Research Reanal
ysis datasets which are provided in 6-hour intervals (0:00, 06:00, 12:00, and 18:00 UTC) with a horizon tal resolution of 2.5 by 2.5 degrees. The measurements of surface pressure were available at the FTS sta tion, which we have used for calculating $XCO_2$ and $XCH_4$. Above the tropopause height, GFIT apriori p rofiles were utilized to extrapolate the aircraft profile. Eventually, the completed aircraft profiles based on those assumptions were transformed into a total column $XCO_2$ and $XCH_4$ by pressure weighting func
tions. For this comparison, we considered only the FTS averaged $XCO_2$ and $XCH_4$ retrieval values for t he corresponding aircraft measurement time. Details about the aircraft $XCO_2$ and $XCH_4$ values during a scending and descending aircraft flight duration and the corresponding FTS averaged $XCO_2$ and $XCH_4$ r etrieval values are also provided in Table 3. Note that the vertically resolved FTS column-averaging ker nels were taken into account for smoothing the aircraft profiles. The $XCO_2$ and $XCH_4$ for the aircraft in
situ profile weighted by the column averaging kernel a (Rodgers and Connor, 2003) is computed as foll ows:

$$X^{in-situ} = X^a + \sum_j h_j a_j (t_{in-situ} - t_a)_j$$

Where $X^a$ is the column-averaged dry air mole fraction for the apriori profile $t_a$ ($CO_2$ or $CH_4$), $t_{in-situ}$ is the aircraft profile and $h_j$ is the pressure weighting function.

We estimated the uncertainty of the $XCO_2$ and $XCH_4$ columns derived from the extended aircraft
profiles by assigning uncertainties. Uncertainty at the surface was assumed to be same as the uncertainty in the lowest measurements. For the stratosphere, we used the method suggested by Wunch et al. (2010). This method shifts the stratospheric values up and down by 1 km to calculate the difference in the total column, which is used as an estimate of the uncertainty in the location of the tropopause and therefore for the stratospheric contribution. We estimated the stratospheric errors in
aircraft integrated amount of $XCO_2$ and $XCH_4$ by shifting the apriori profile by 1 km (Ohyama et al. 2015). For KORUS-AQ, it was found to be 0.42 ppm for $XCO_2$ and 13.26 ppb for $XCH_4$.

For NASA's DC8 measurements, the in-situ profiles covered the altitude range of approximately 0.17 to 9.0 km, in-situ surface data were utilized near the surface to complement the aircraft profiles and extended the aircraft ceiling point of measurements to the tropopause which is estimated by NCEP to be
at 139.0 hPa. Figure 9 illustrates the results of $XCO_2$ and $XCH_4$ comparisons between the aircraft observation and TCCON site data. In this plot, blue represents the KORUS-AQ campaign, whereas green indicates the NIMS campaign. KORUS-AQ data lie on the best line which is derived using TCCON stations where aircraft profiles are available. This shows that TCCON Anmyeondo data is consistent with other TCCON stations.

## 2.7 Comparison with OCO-2 measurements

The Orbiting Carbon Observatory-2 (OCO-2) is NASA's first Earth-orbiting satellite dedicated to greenhouse gas measurement, it was successfully launched on July 2, 2014 into low-Earth orbit. It is devoted to observing atmospheric carbon dioxide ($CO_2$) to provide improved insight into the carbon
cycle. The primary mission is to measure carbon dioxide with high precision and accuracy in order to characterize its sources and sinks at different spatial and temporal scales (Boland et al., 2009; Crisp, 2008, 2015). The instrument measures the near infrared spectra (NIR) of sunlight reflected off the Earth's surface. Atmospheric abundances of carbon dioxide and related atmospheric parameters are retrieved from the spectra in nadir, sun glint and target modes. Detailed information about the
instrument is available in, for example (Connor et al., 2008; O'Dell et al., 2012). In this work, we used the OCO-2 version 7Br bias corrected data. The comparisons are discussed in section 3.3.

## 3 Results and discussion

## 3.1 Time series of g-b FTS $XCO_2$, seasonal and annual cycle

The time series of $XCO_2$ along with retrievals of other trace gases such as XCO and $XCH_4$ from g-b FTS is presented in Figure 10 (panel a-c) for the period from February 2014 to November 2017. In these time series plots, each marker represents a single retrieval, and the fitting curves of the retrieved values are also depicted (red solid line). We show the seasonal cycle of $XCO_2$, XCO, and $XCH_4$ in the time series using a fitting procedure described by Thoning et al. (1989). Standard deviations of the differences between the retrieved values and the fitting curves are 1.64 ppm, 11.34 ppb, and 10.1 ppb for $XCO_2$, XCO, and $XCH_4$, respectively. It is evident that all species have a seasonal cycle feature. Year to year variability of $XCO_2$ is highest in spring and lowest during the growing season in June to September. Moreover, the behavior of the seasonal cycle of $XCO_2$ at our site was compared with that of $XCO_2$ at Saga, Japan, which is discussed in a later section. The atmospheric increase of $XCO_2$ from 2015 to 2016 was 3.65 ppm, which is larger than the increase from 2014 to 2015. For the case of $XCH_4$, its increase from 2015 to 2016 was 0.02 ppm, which is higher than the increase from 2014 to 2015, whereas in XCO the rate of increment from year to year was found to be slightly decreased (see Table 4).

The seasonal and annual cycles of $XCO_2$ derived from the g-b FTS were compared with in-situ tower observations of $CO_2$ over the Anmyeondo station, which are presented in Figure 11. Regarding in-situ data, samples were collected using flasks and analysed using non-dispersive infrared (NDIR) spectroscopy at the altitude of 77 meters above sea level (details about in situ data are available at http://ds.data.jma.go.jp/jmd/wdcgg/). Nearly 97 % of in-situ data in Figure 11 were taken during day time between 04:00 – 08:40 UTC (13:00 – 17:40 Korea Standard Time (KST)) so that the early morning and night time enhancements of $CO_2$ were mostly excluded. In-situ $CO_2$ monthly means are generated by first averaging all valid event measurements with a unique sample date and time. The values are then extracted at weekly intervals from a smooth curve (Thoning et al., 1989) fitted to the averaged data and then these weekly values are averaged for each month. As can be seen in Figure 10, the overall patterns of seasonal and annual cycle of FTS $XCO_2$ tend to be similar with those of in-situ tower $CO_2$.

## 3.2 Comparison of Anmyeondo $XCO_2$ with nearby TCCON station

In Figure 12, we present the comparison of our FTS $XCO_2$ data with a similar ground-based high resolution TCCON FTS observation at Saga station (33.26 N, 130.29 E) in Japan, which is the closest TCCON station to our site. Among nearby TCCON stations, Rikubetsu, Tsukuba, and Saga are located in Japan (Morino et al., 2011, Ohyama et al., 2009, 2015) and Hefei is located in China (Wang et al., 2017). To demonstrate the comparison between them, we have shown the daily averaged $XCO_2$ of two stations during the period of 2014 to 2017 in Figure 12. As can be seen, variations of $XCO_2$ at the Saga station agreed well with Anmyeondo station. The daily averaged $XCO_2$ revealed the same seasonal

cycle as that of our station. The lowest $XCO_2$ appeared in late summer (August and September), and the
highest value was in spring (April).

Ohyama et al., (2015) studied the time series of $XCO_2$ at Saga, Japan during the period from July 2011 to December 2014. They showed seasonal and interannual variations. The peak-to-peak seasonal amplitude of $XCO_2$ was 6.9 ppm over Saga during July 2011 and December 2014, with a seasonal maximum and minimum in the average seasonal cycle during May and September, respectively. In recent findings of Wang et al. (2017), the g-b FTS temporal distributions of $XCO_2$ at Hefei, China were reported. The FTS observations in 2014 to 2016 had a clear and similar seasonal cycle, i.e. $XCO_2$ reaches a minimum in late summer, and then slowly increases to the highest value in spring. The daily average of $XCO_2$ ranges from 392.33 $\pm$ 0.86 to 411.62 $\pm$ 0.90 ppm, and the monthly average value shows a seasonal amplitude of 8.31 and 13.56 ppm from 2014 to 2015 and from 2015 to 2016, respectively. The seasonal cycle was mainly driven by large scale (hemispheric) biosphere–atmosphere exchange. Butz et al., (2011) reported that the observations from GOSAT and the co-located ground-based measurements agreed well in capturing the seasonal cycle of $XCO_2$ with the late summer minimum and the spring maximum for four TCCON stations (Bialystok, Orleans, Park Falls, and Lamont) in the Northern Hemisphere. We infer that the variation of $XCO_2$ over Anmyeondo station is in harmony with the variation pattern in mid-latitude Northern Hemisphere.

## 3.3 Comparison of $XCO_2$ between the g-b FTS and OCO-2

In this section, we present a comparison of $XCO_2$ between the g-b FTS and OCO-2 version 7Br data (bias corrected data) over Anmyeondo station during the period between 2014 and 2017. For making a direct comparison of the g-b FTS measurements against OCO-2, we applied the spatial coincidence criteria for the OCO-2 data within 3° latitude/longitude of the FTS station, as well as setting up a time window of 3 hours (maximum 3 hours mismatch between satellite and g-b FTS observations). Based on the coincidence criteria, we obtained 13 coincident measurements, which were not sufficient to infer a robust conclusion, but do provide a preliminary result. The comparison of the time series of $XCO_2$ concentrations derived from the g-b FTS and OCO-2 on daily median basis is demonstrated during the measurement period between 2014 and 2017, depicted in Figure 13. As can be seen in the plot, the g-b FTS measurement exhibits some gaps which occurred due to bad weather conditions, instrument failures, and absences of an instrument operator. In the present analysis, the $XCO_2$ concentrations from FTS were considered only when retrieval error was below 1.50 ppm (not shown), which is the sum of all error components such as laser sampling error, zero level offsets, ILS error, smoothing error, atmospheric a-priori temperature, atmospheric a-priori pressure, surface pressure, and random noise. Wunch et al. (2016) reported that the comparison of $XCO_2$ derived from the OCO-2 version 7Br data against co-located ground-based TCCON data that indicates the median differences between the OCO-2

and TCCON data were less than 0.50 ppm, and corresponding RMS differences of less than 1.50 ppm. The overall results of our comparisons were comparable with the report of Wunch et al. (2016). The OCO-2 product of $XCO_2$ was biased (satellite minus g-b FTS) with respect to the g-b FTS, which was slightly higher by 0.18 ppm with a standard deviation of 1.19 ppm, a corresponding RMS difference of 1.16 ppm. This bias could be attributed to the instrument uncertainty. In addition to that, we also obtained a strong correlation between the two datasets, which was quantified as a correlation coefficient of 0.94 (see Table 5 and Figure 13).

Both measurements capture the seasonal variability of $XCO_2$. As can be seen clearly from the temporal distribution of FTS $XCO_2$, the maximum and minimum values are discernible in spring and late summer seasons, respectively. The mean values in spring and summer were 402.72 and 396.92 ppm, respectively (see Table 6). This is because the seasonal variation of $XCO_2$ is most likely to be controlled by the imbalance of the terrestrial ecosystem exchange, and this could explain the larger $XCO_2$ values in the northern hemisphere in late April (Schneising et al. 2008, and references therein). The minimum value of $XCO_2$ occurs in August, which is most likely due to uptake of carbon into the biosphere associated with the period of plant growth. Furthermore, both instruments showed high standard deviations during summer, about 3.28 ppm in FTS and 3.77 ppm in OCO-2, and this suggests that the variability reflects strong sources and sink signals.

## 4 Conclusions

Monitoring of greenhouse gases is an essential issue in the context of global climate change. Accurate and precise continuous long-term measurements of greenhouse gases (GHGs) are substantial for investigating their sources and sinks. Today, several remote sensing instruments operated on different platforms are dedicated for measuring GHGs. Total column measurements of greenhouse gases such as $XCO_2$, $XCH_4$, $XH_2O$, $XN_2O$ have been made using the g-b FTS at the Anmyeondo station since 2013. In this work, we focused on the measurements taken during the period of February 2014 to November 2017. The instrument has been operated in a semi-automated mode since then. The FTS instrument has been stable during the whole measurement period. Regular instrument alignment checks using the HCl cell measurements are performed. The TCCON standard GGG2014 retrieval software was used to retrieve $XCO_2$, $XCO$, and others GHG gases from the g-b FTS spectra.

In this work, the g-b FTS retrieval of $XCO_2$ and $XCH_4$ were compared with aircraft measurements that were conducted over Anmyeondo station on 22 May 2016, 29 October and 12 November, 2017. The mean absolute difference between FTS and aircraft $XCO_2$ were found to be -1.109 ± 0.802 ppm, corresponding to a mean relative difference of -0.273 ± 0.198 % for $XCO_2$, while the mean absolute difference for $XCH_4$ is 0.007 ± 0.0096 ppm, corresponding to a mean relative difference of 0.377 ±

0.518 %. These differences appeared in both species and were consistent with the combined instrument errors. The preliminary comparison results of $XCO_2$ between FTS and OCO-2 were also presented over the Anmyeondo station. The mean absolute difference of $XCO_2$ between FTS and OCO-2 was calculated on daily median basis, and it was estimated to be 0.18 ppm with a standard deviation of 0.19 with respect to the g-b FTS. This bias could be attributed to instrument uncertainty. Based on the seasonal cycle comparison, both the g-b FTS and OCO-2 showed a consistent pattern in capturing the seasonal variability of $XCO_2$, with maximum in spring and minimum in summer.

## 5 Acknowledgements

This research was supported by the Research and Development for KMA Weather, Climate, and Earth system Services (NIMS-2016-3100). D.W.T.G and V.A.V. would like to acknowledge financial support from the Australian Research Council (ARC) for TCCON activities (DP160101598, DP140101552, DP110103118). We are grateful for the Jeong-Hoo Park in Park National Institute of Environmental Research and Ryan Bennett in NASA Langley Research Center and NASA's KORUS campaign team. Many thanks goes to Dr. Haeyoung Lee for accessing in-situ surface data of the Anmyeondo station and Dr. Kei Shiomi and the team as well for the provision of Saga FTS data. We thank the OCO-2 and GOSAT science teams for the delivery of data. Finally, we strongly appreciate the two anonymous reviewers who helped us to improve this manuscript well.

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

# Tables

**Table 1.** Measurement setting for the Anmyeondo g-b FTS spectrometer of the Bruker 125HR model.

| Item | Setting |
|------|---------|
| Aperture (field stop) | 0.8 mm |
| Detectors | RT-Si Diode DC, |
| | RT-InGaAs DC |
| Beamsplitters | $CaF_2$ |
| Scanner velocity | 10 kHz |
| Low pass filter | 10 kHz |
| High folding limit | 15798.007 |
| Spectral Resolution | 0.02 $cm^{-1}$ |
| Optical path difference | 45 cm |
| Acquisition mode | Single sided, forward backward |
| Sample scan | 2 scans, forward, backward |
| Sample scan time | ~110 s |

**Table 2.** Spectral windows used for the retrievals of the columns of $CO_2$ and $O_2$

| Gas | Center of spectral window ($cm^{-1}$) | Width ($cm^{-1}$) | Interfering gas |
|-----|-----|-----|-----|
| $O_2$ | 7885.0 | 240.0 | $H_2O$, HF, $CO_2$ |
| $CO_2$ | 6220.0 | 80.0 | $H_2O$ ,HDO, $CH_4$ |
| $CO_2$ | 6339.5 | 85.0 | $H_2O$ ,HDO |

**Table 3.** Summary of the column averaged dry-air mole fractions obtained during the inter-comparison between the in-situ instrument on board the aircrafts and the g-b FTS at the Anmyeondo station. A and D represent ascending and descending, respectively. Note that FTS values given below are without TCCON common scale factor and FTS column averaging kernels are applied to the aircraft data.

| Date of measurements (hours in KST) | Aircraft NIMS $XCO_2$ (ppm) | g-b FTS $XCO_2$ (ppm) | Aircraft NIMS $XCH_4$ (ppm) | g-b FTS $XCH_4$ (ppm) |
|-----|-----|-----|-----|-----|
| **2017-10-29** | | | | |
| 09:59:16-10:31:08 (A) | 409.152 | 404.242 | 1.8900 | 1.8460 |
| 10:31:09-11:03:24 (D) | 409.336 | 403.877 | 1.8854 | 1.8454 |
| 12:58:58-13:37:07 (A) | 407.011 | 401.051 | 1.8562 | 1.8265 |
| 13:37:07-14:19:40 (D) | 406.898 | 400.537 | 1.8720 | 1.8249 |
| **2017-11-12** | | | | |
| 11:12:20-11:38:01 (A) | 406.541 | 401.839 | 1.8513 | 1.8221 |
| 11:38:02-12:13:00 (D) | 406.839 | 401.930 | 1.8512 | 1.8220 |
| 14:14:46-14:45:55 (A) | 406.517 | 401.592 | 1.8479 | 1.8201 |
| 14:45:56-15:23:47 (D) | 407.628 | 401.473 | 1.8504 | 1.8191 |
| **Mean ± std** | **407.491 ±1.137** | **402.068± 1.311** | **1.8630± 0.0170** | **1.8283± 0.011** |
| | **KORUS** | **TCCON** | **KORUS** | **TCCON** |
| 2016-05-22 | 405.80 ± 0.42 | 401.91 ± 0.57 | 1.8641± 0.0132 | 1.8100±0.002 |

**Table 4.** Annual mean of $XCO_2$, XCO, and $XCH_4$ from Anmyeondo g-b FTS from February 2014 to November 2017 is given.

| Gases | Annual mean ± standard deviation | | | |
|---|---|---|---|---|
| | 2014 | 2015 | 2016 | 2017 |
| $XCO_2$ (ppm) | 396.91 ± 2.55 | 399.32 ± 2.96 | 402.97 ± 2.74 | 406.04 ± 2.38 |
| XCO (ppb) | 99.42 ± 14.71 | 102.73 ± 14.91 | 105.39 ± 10.68 | 100.14 ± 10.3 |
| $XCH_4$ (ppm) | 1.837 ± 0.014 | 1.844 ± 0.015 | 1.864 ± 0.015 | 1.859 ± 0.013 |

**Table 5.** Summary of the statistics of $XCO_2$ comparisons between OCO-2 and the g-b FTS from 2014 to 2017 are presented. N - coincident number of data, R - Pearson correlation coefficient, RMS - Root Mean Squares differences.

| N | Mean Absolute.diff. (ppm) | Mean Relative diff (%) | R | RMS (ppm) |
|---|---|---|---|---|
| 13 | 0.18±1.19 | 0.04±0.29 | 0.94 | 1.16 |

**Table 6.** Seasonal mean and standard deviations of $XCO_2$ from the g-b FTS and OCO-2 in the period between 2014 and 2016 are given below.

| Season | g-b FTS $XCO_2$ mean ± std (ppm) | OCO-2 $XCO_2$ mean ± std (ppm) |
|---|---|---|
| Winter | 401.52 ± 0.85 | 402.67 ± 2.67 |
| Spring | 402.72 ± 2.79 | 403.96 ± 2.77 |
| Summer | 396.92 ± 3.28 | 399.68 ± 3.77 |
| Autumn | 398.01 ± 2.83 | 398.48 ± 2.41 |

# Figures

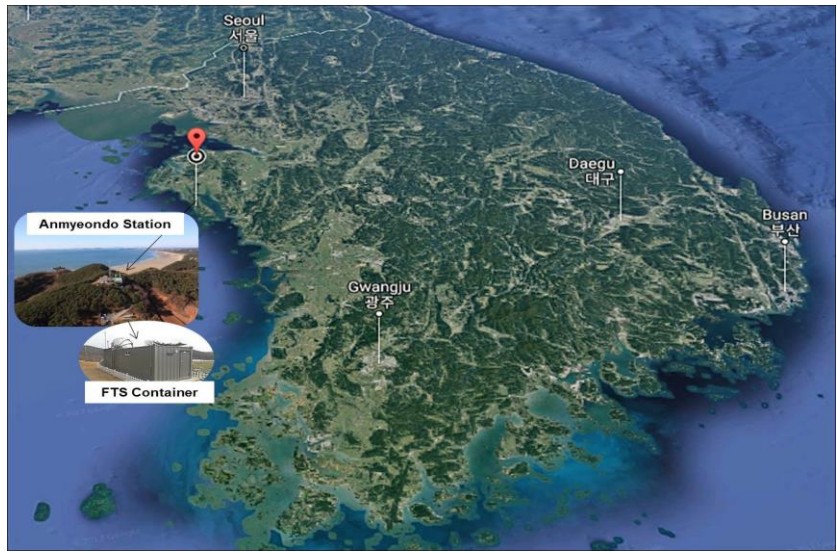

**Figure 1.** Anmyeodo (AMY) g-b FTS station

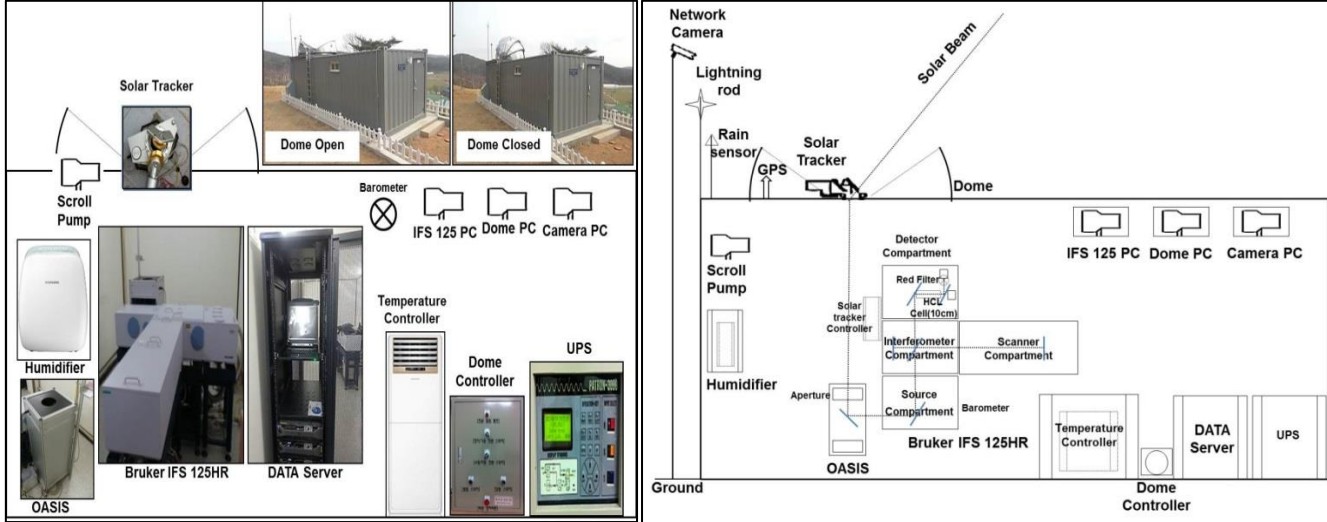

**Figure 2.** Photographs of the automated FTS laboratory. The Bruker Solar Tracker type A547 is mounted in the custom made dome. A servo controlled solar tracker directs the solar beam through a $CaF_2$ window to the FTS (125HR) in the laboratory. The server computer is used for data acquisition. PC1 and PC2 are used for controlling the spectrometer, solar tracker, dome, camera, pump, GPS satellite time, and humidity sensor.

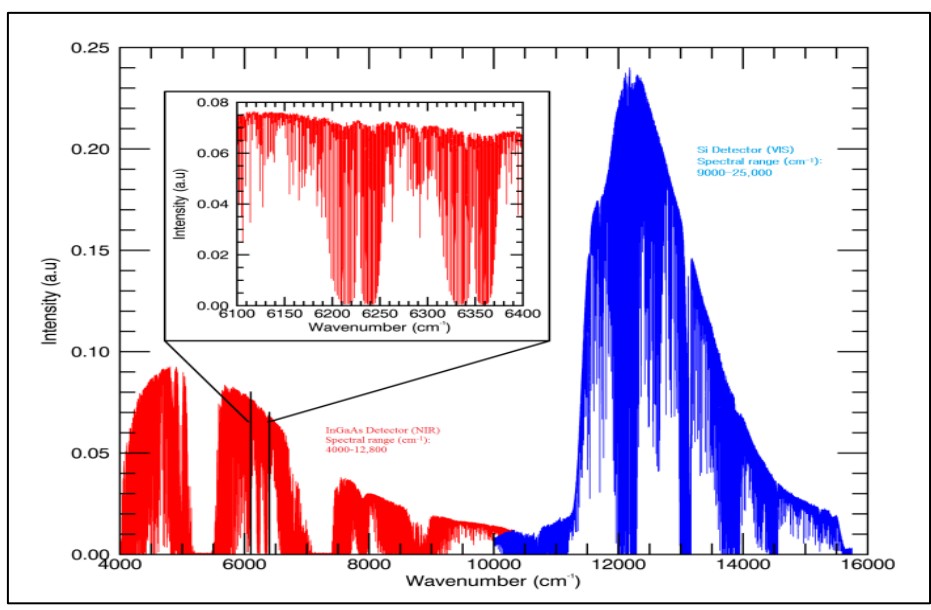

**Figure 3.** Single spectrum recorded on 4 October 2014 with a resolution of 0.02 cm$^{-1}$. A typical example for the spectrum of -XCO$_2$ is shown in the inset.

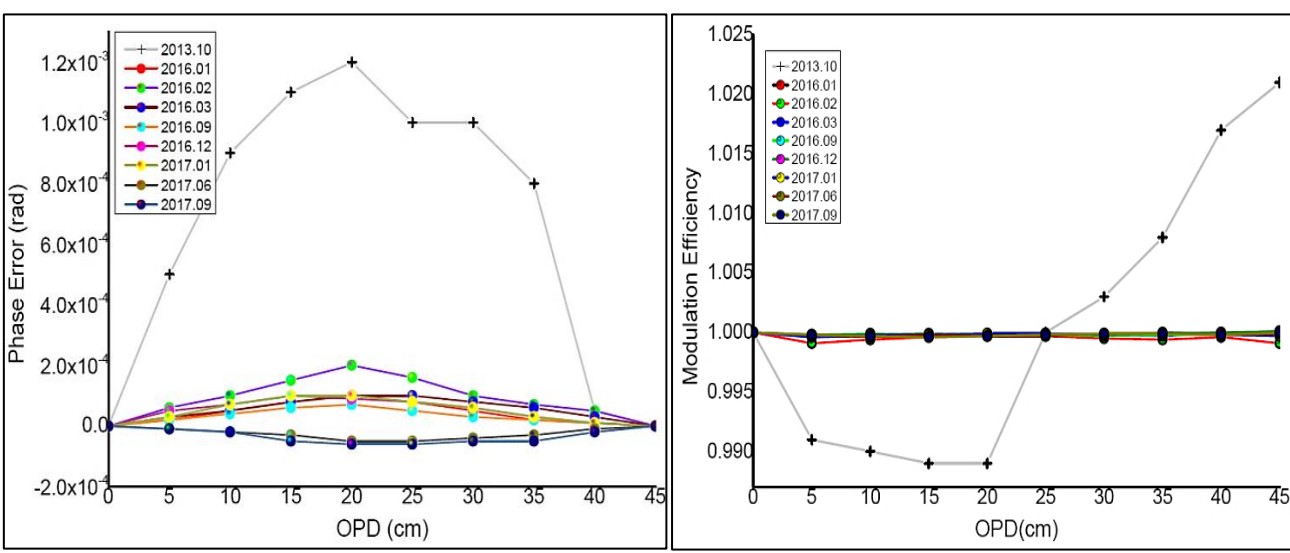

**Figure 4.** Modulation efficiency (right panel) and Phase error (rad) (left panel) of HCl measurements from the g-b FTS are displayed in the period from October 2013 to September 2017. Resolution = 0.02 cm$^{-1}$, Aperture = 0.8 mm.

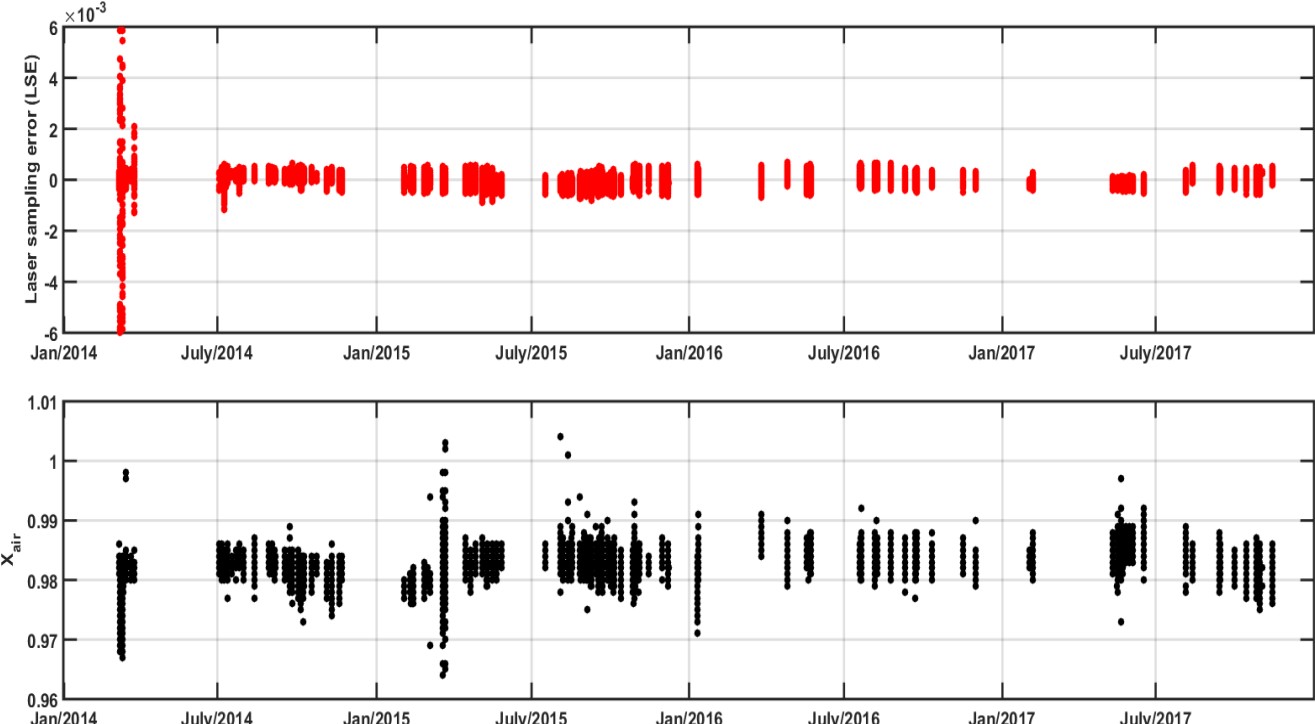

**Figure 5.** Time series of LSE (top panel) and $X_{air}$ (bottom panel) from the g-b FTS during 2014- 2017 is shown. Each marker represents a single measurement.

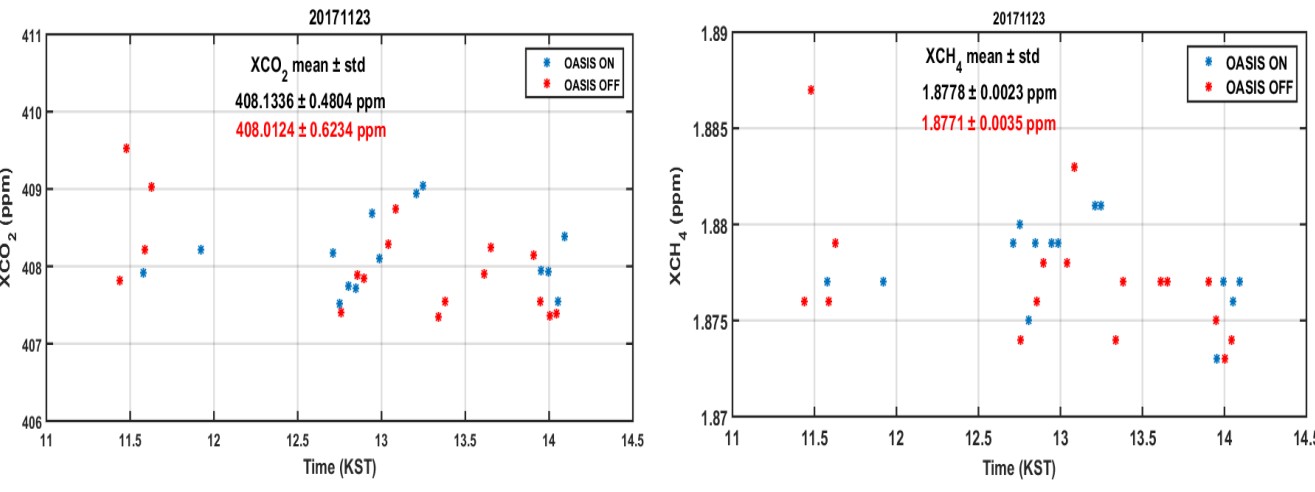

 **Figure 6.** G-b FTS $XCO_2$ (left panel) and $XCH_4$ (right panel) values as function of time in KST (Korean Standard time, UTC+9) taken October 23, 2017 with OASIS system on (operating) and off (without operating) positions are shown. Each marker represents a single measurement.

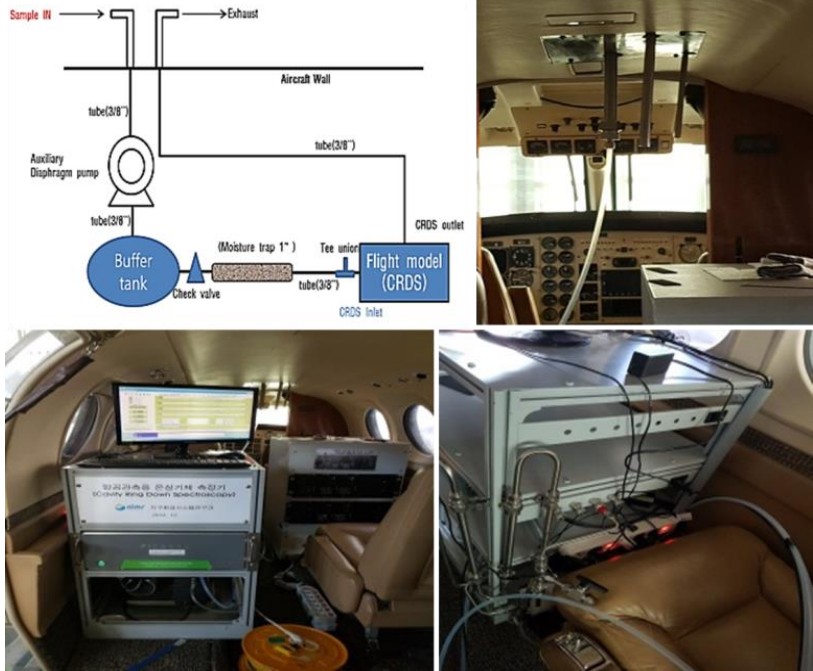

**Figure 7.** NIMS CRDS instrument on board the King Air 90C.

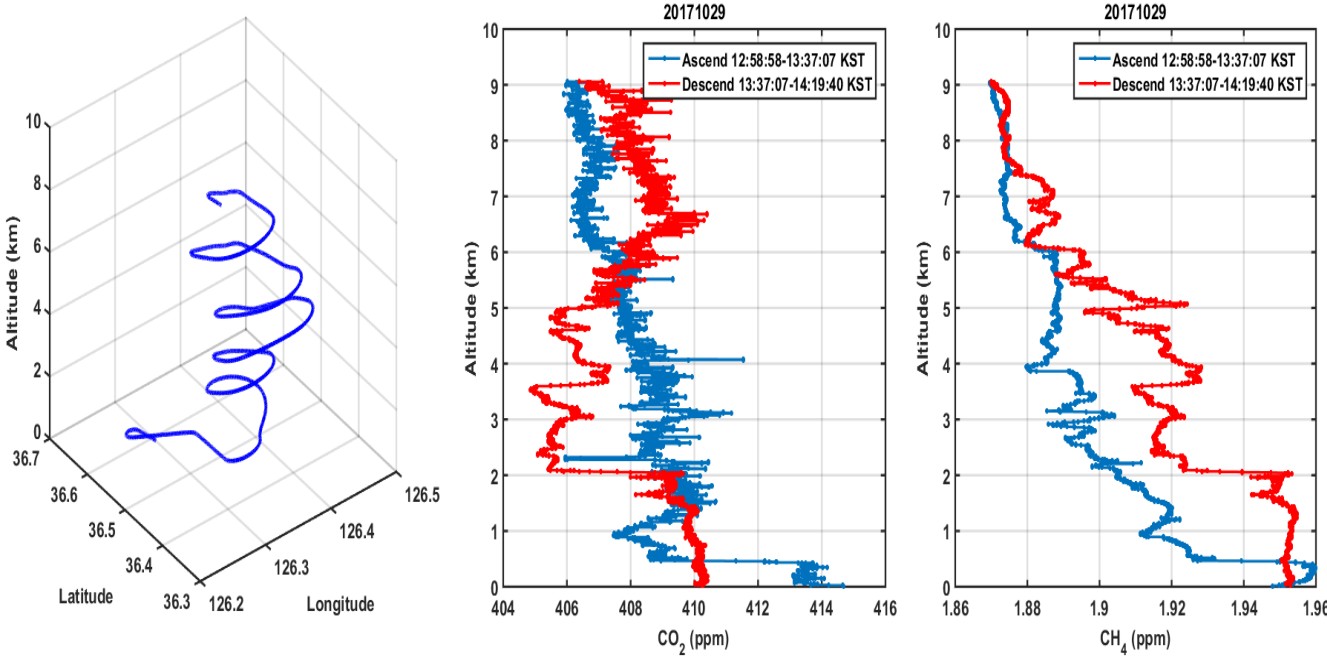

**Figure 8.** Typical flight path (left panel), $CO_2$ (middle panel) and $CH_4$ (right panel) VMR profiles during ascent and descent of the aircraft over Anmyeondo on October 29, 2017 are shown.

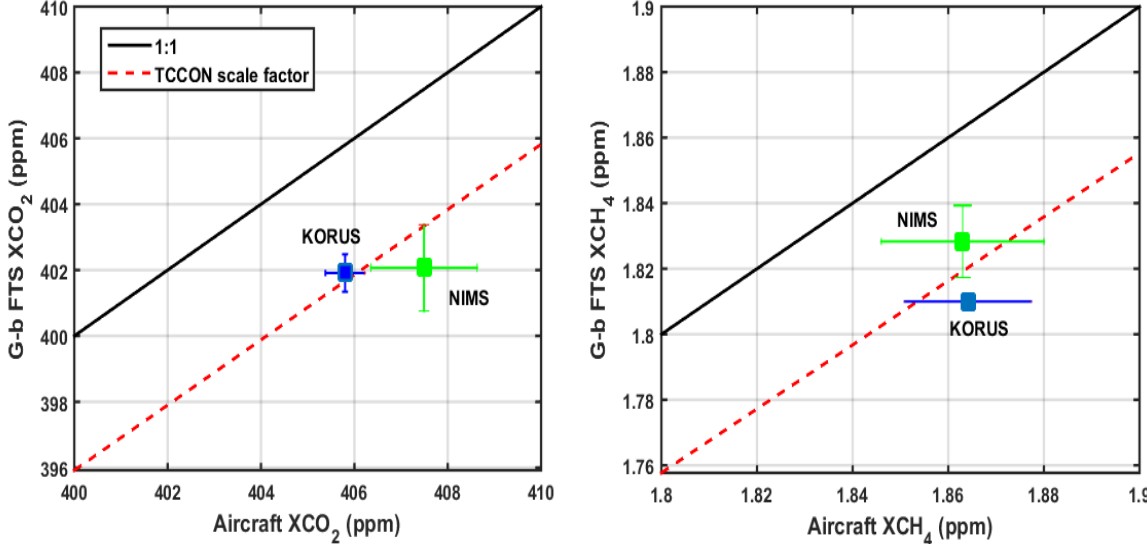

**Figure 9.** The comparisons of $XCO_2$ and $XCH_4$ between the aircraft observation and g-b FTS data over Anmyeondo station are shown. The blue square symbol represents for the aircraft campaign conducted by KORUS-AQ (May, 2016), whereas green square symbol indicates for the aircraft campaign operated by NIMS (2017). Note that FTS values shown in the figure are after removing TCCON common scale factor.

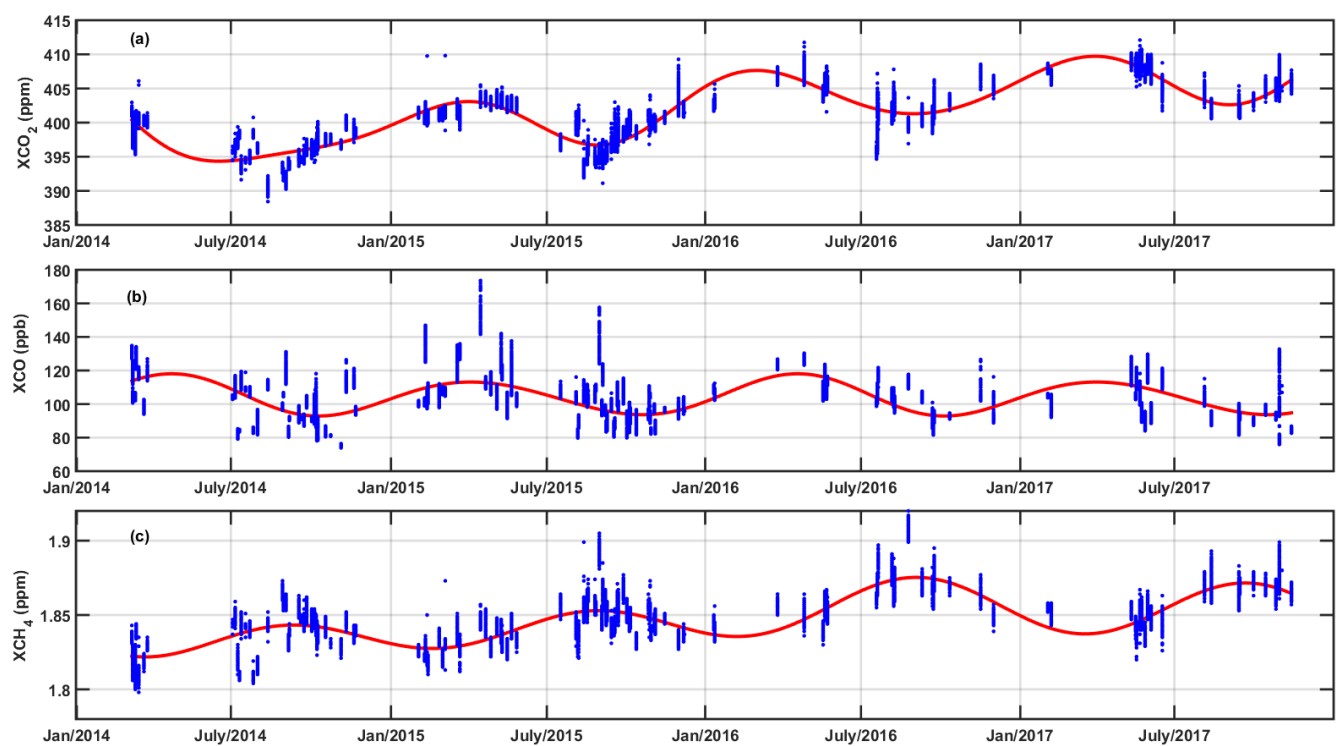

**Figure 10.** Time series of $XCO_2$, XCO, and $XCH_4$ from top to bottom panels (a-c), respectively in the period between February 2014 and November 2017 is given. Each marker indicates a single retrieval. Fitting curves (red solid lines) are also displayed.

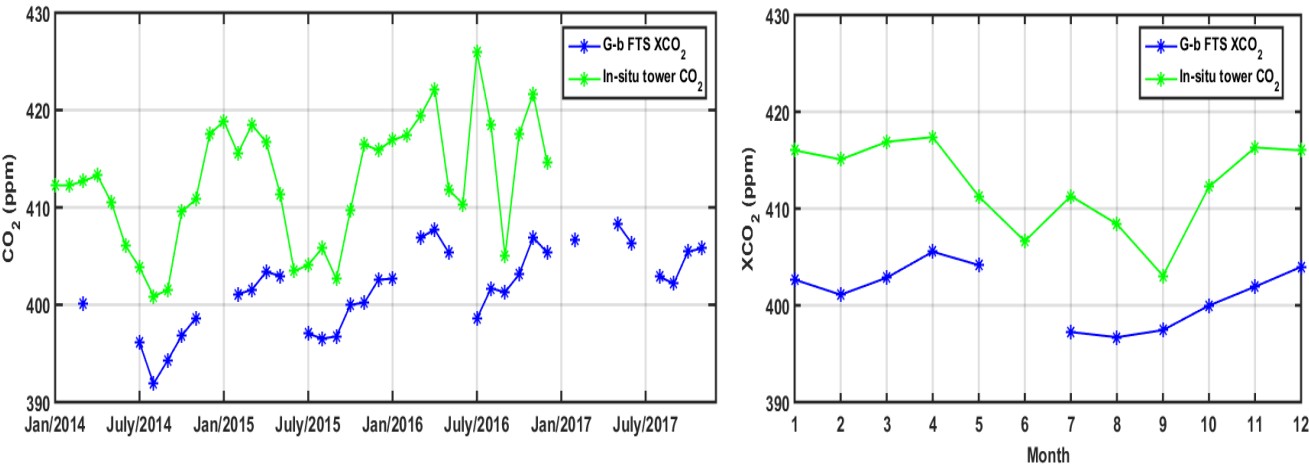

**Figure 11.** Left panel shows the time series of FTS $XCO_2$ and in-situ tower $CO_2$ on monthly mean basis, whereas right panel depicts annual cycle (2014-2016).

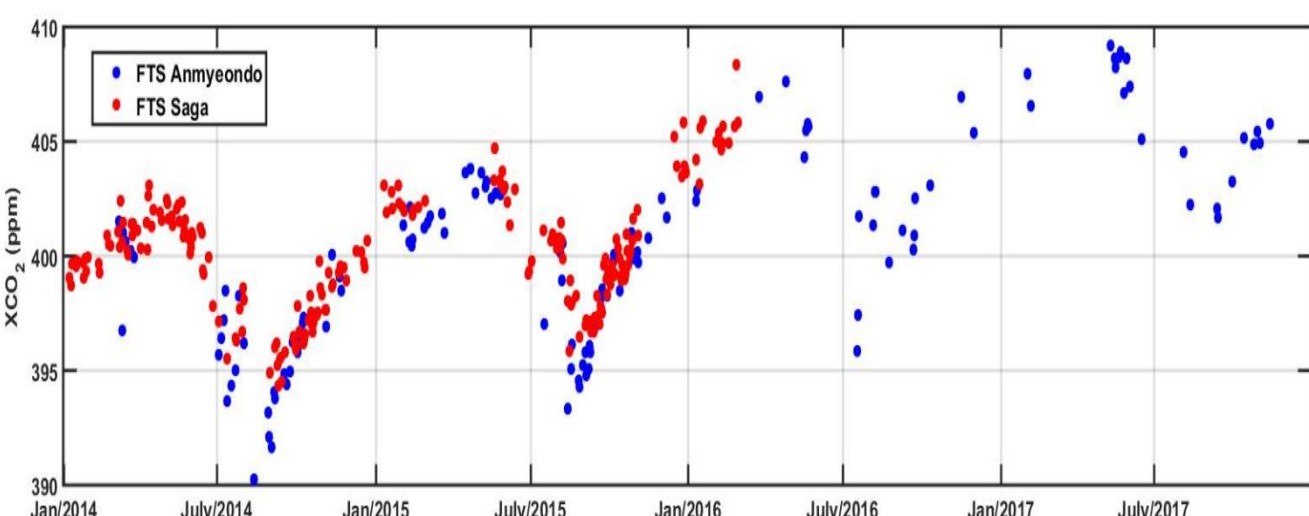

**Figure 12.** Time series of daily averaged $XCO_2$ retrieval from Anmyeondo FTS and Saga FTS in the period of February 2014 to November 2017 is depicted.

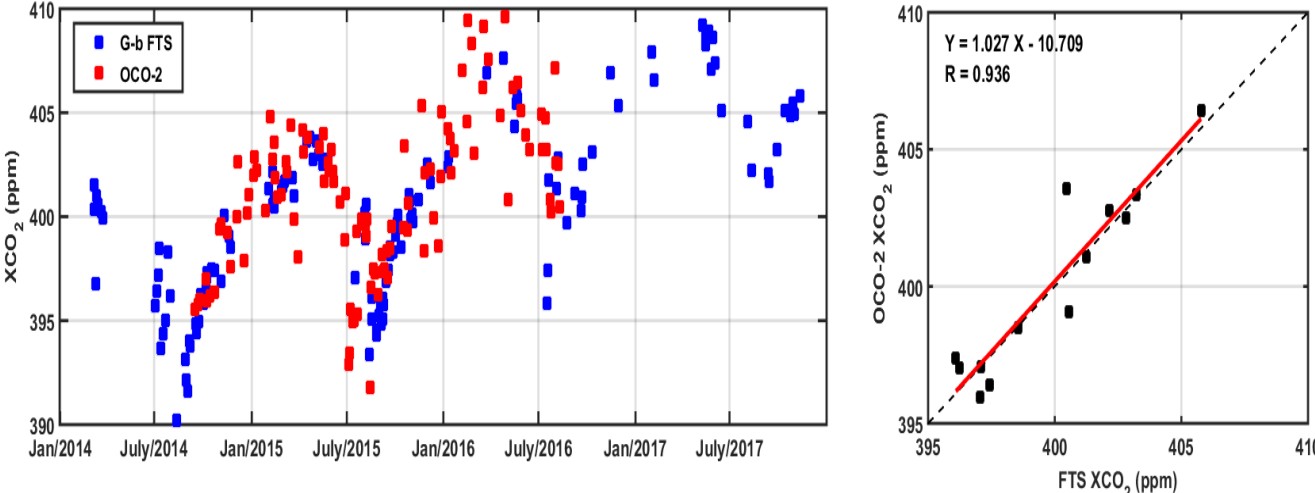

**Figure 13.** Left panel: The time series of $XCO_2$ from the g-b FTS (blue squares) and OCO-2 (red squares) over the Anmyeondo station from February 2014 to November 2017 are shown. Right panel: The linear regression curve between FTS and OCO-2 is shown. All results are given on a daily medians basis.