# Peer review of "Characteristics of Greenhouse Gas Concentrations Derived from Ground-based FTS Spectra at Anmyeondo, Korea"

_Atmospheric Measurement Techniques, 2017_

## Referee Comment (RC1) · T. Blumenstock (Referee) · 24 May 2017

General comments:

The authors present a new TCCON site in Korea. This paper characterizes the instrumentation and gives an example of its application: inter-comparison with OCO-2 satellite data. This site really fills a gap in the existing TCCON network and will be very useful to assessing sink and sources of GHGs. The data and also the comparison with OCO-2 data are of good quality.

The subject is appropriate for publication in AMT. The paper is well written and I recommend publication after major revisions, in particular a more comprehensive description

of the variable aperture system OASIS system is needed.

Major comment:

A specific feature of the described instrumentation is the so called OASIS (Operational Automatic System for Intensity of Sunray) system. While analog systems are used in active remote sensing systems, for example laser output control in LIDAR systems, an intensity control of passive systems is typically not used.

In the TCCON network the variability of the DC signal is used to quality check and correct the recorded interferograms and resulting spectra. Since you remove this signal you cannot apply this kind of quality check anymore. Do you record and use the actual setting of the aperture to do so?

If the motivation to introduce such a system is to limit the intensity to avoid non-linear response a smaller constant aperture or smaller preamp gain or a smaller sensitivity of the detector might be more appropriate. Would you please add a statement for the motivation to add this system. Or a comparison of XCO2 time series recorded with and without OASIS system which might demonstrate the difference, for example in terms of signal to noise ratio.

Where is the variable aperture positioned? Is it in the parallel or focused beam? Did you check the influence on the ILS (Instrumental Line Shape) due to the variable aperture while scanning?

I assume a lamp was used and hence the OASIS system was not active while performing cell measurements. Cell measurements using the sun as source might be an option to check the ILS while the OASIS system is active. Or, if the HCl lines in the atmospheric spectrum are covered by interfering species you might do cell measurements with the lamp using different fixed aperture settings to check the influence of the OASIS system on the ILS.

How does your system and its influence on the ILS compares with the results of the

recent paper by Sun et al, AMT, 2017 on the 'Sensitivity of instrumental line shape monitoring for the ground-based high-resolution FTIR spectrometer with respect to different optical attenuators'?

While most of the site complies with the TCCON standard setup the OASIS system does not. Therefore a more detailed description is needed as well as a discussion on its influence on the ILS.

Specific comments:

- In Chapter 3.1 the time series of the $O_2$ columns is compared with atmospheric pressure. Therefore including surface pressure in Fig. 8 might support your statement.

- The errors are shown in Fig. 9. How is the error calculated and which sources of errors are included?

- Can you specify 'regular cell measurements'?

Technical comments:

- p.1 + 12: were generally agreed => generally agreed

- p.2: space born => space borne

- p.3: area is; => area is:

- p.4: with oil-free => with oil-free pump

- p.5: beamspliters =< beamsplitters

- p. 7: to these derived => to those derived (?)

- p. 9:

- orbit, devoted => orbit. It is devoted

- can available => is available

- p.10: are varied => varied

- p.11:

- over the land => over land

- square => squares

- p.12:

- and this suggesting => suggesting

- new page within Table 4

p.13:

- the source and sink of them. => their sources and sinks.

- outcome this => outcome of this

- Is '... to be withered that turns out to be weak photosynthesis ... a grammatically correct sentence?

Since I'm not a native speaker I would like to suggest to ask a native speaker of the author list to decide on the technical suggestions.

———————————————————

---

## Author Comment (AC1) · 13 Jul 2017

Response to interactive comment of Referee on "Interactive comment on "Characteristics of the Greenhouse Gas Concentration Derived from the Ground-based FTS Spectra at Anmyeondo, Korea" by Young-Suk Oh et al. T. Blumenstock (Referee)

General comments: The authors present a new TCCON site in Korea. This paper characterizes the instrumentation and gives an example of its application: inter-comparison with OCO-2 satellite data. This site really fills a gap in the existing TCCON network and will be very useful to assessing sink and sources of GHGs. The data and also

the comparison with OCO-2 data are of good quality. The subject is appropriate for publication in AMT. The paper is well written and I recommend publication after major revisions, in particular a more comprehensive description. Response First of all, we would like to strongly appreciate referee's very constructive and valuable comments on the manuscript. We have tried to address all the issues (major and minor comments) raised on this paper one by one. The referee makes strong comments to give more details about OASIS and its influence on ILS (Instrumental Line Shape) by including some illustrations so that we added some more brief on it. Our replies and respective changes are described below. Technical comments regarding spellings and grammatical errors are corrected in the final version of the manuscript.

Major comment: A specific feature of the described instrumentation is the so called OASIS (Operational Automatic System for Intensity of Sunray) system. While analog systems are used in active remote sensing systems, for example laser output control in LIDAR systems, an intensity control of passive systems is typically not used. In the TCCON network the variability of the DC signal is used to quality check and correct the recorded interferograms and resulting spectra. Since you remove this signal you cannot apply this kind of quality check anymore. Do you record and use the actual setting of the aperture to do so? Response In addition to OASIS system, we have also simultaneously used the variability of DC signal similar to TCCON network for quality control of the spectra and its retrieval results. Yes, the spectra are recorded with the actual setting of the aperture having a diameter of 0.8 mm throughout the observation period.

If the motivation to introduce such a system is to limit the intensity to avoid non-linear response a smaller constant aperture or smaller preamp gain or a smaller sensitivity of the detector might be more appropriate. Would you please add a statement for the motivation to add this system? Or a comparison of XCO2 time series recorded with and without OASIS system which might demonstrate the difference, for example in terms of signal to noise ratio. Response The OASIS system is developed for improving the

quality of the spectra. To ensure the quality of the spectra, this system will be useful for minimizing the noise that induced in the spectra due to rapid intensity fluctuations of the incoming solar radiation that reaches to the instrument. This rapid intensity fluctuations are be occurred in the presence of clouds, aerosol loading etc. along the path of incoming radiation within the instrument field of view. To minimize this intensity fluctuations due to the changing weather conditions, OASIS system regulates in such a way that by varying the aperture size at the source compartment based on the signals from photon sensor which depends on the levels of incoming sunlight intensity. Thereby, it avoids non-linear response of a smaller constant aperture or smaller preamp gain or a smaller sensitivity of the detector. In this study, we are not able to show the whole time series of XCO2 without OASIS system during the study period since all spectra that are used for analysis of species are obtained after the OASIS system equipped to our FTS spectrometer. However, for a typical example, we illustrated the time series of XCO2 in both cases. Based on TCCON community suggestions regarding the OASIS system, it would be recommended to use a consistent g-b FTS measurement set up throughout TCCON network so that we plan to fix a constant aperture size at the source compartment during the FTS operation at Anmyeondo station.

Where is the variable aperture positioned? Response A variable aperture is placed inside the OASIS system which is at the source compartment.

Is it in the parallel or focused beam? Response It is a focused beam.

Did you check the influence on the ILS (Instrumental Line Shape) due to the variable aperture while scanning? Response Yes, we assessed the influence of ILS due to the variable aperture and the result showed that it has no impact on the ILS.

I assume a lamp was used and hence the OASIS system was not active while performing cell measurements. Cell measurements using the sun as source might be an option to check the ILS while the OASIS system is active. Or, if the HCl lines in the atmospheric spectrum are covered by interfering species you might do cell measurements with the lamp using different fixed aperture settings to check the influence of the OASIS system on the ILS. How does your system and its influence on the ILS compares with the results of the recent paper by Sun et al, AMT, 2017 on the 'Sensitivity of instrumental line shape monitoring for the ground-based high-resolution FTIR spectrometer with respect to different optical attenuators'? While most of the site complies with the TCCON standard setup the OASIS system does not. Therefore a more detailed description is needed as well as a discussion on its influence on the ILS. Response We have carried out experiments to investigate the influences of ILS due to the presence of OASIS system, and then considered HCl cell measurements using sun as source while OASIS system active and tungsten lamp as a source while OASIS inactive. The result confirmed that the ILS was not affected by the variable aperture during the operation of OASIS system. Sun et al. (2017) reported the detailed characteristics of the ILS with respect to applications of different optical attenuators to FTIR spectrometers within the TCCON and NDACC networks. They used both lamp and sun cell measurements which were conducted after the insertion of five different attenuators in front of and behind the interferometer. In Sun et al. (2017) paper, the ILS result was indicated by considering optical attenuator no .1 which is in good agreement with our findings.

Specific comments: - In Chapter 3.1 the time series of the O2 columns is compared with atmospheric pressure. Therefore including surface pressure in Fig. 8 might support your statement. Response The time series of surface pressure is included in bottom panel of Fig. 8 and compared with the time series of O2 column.

- The errors are shown in Fig. 9. How is the error calculated and which sources of errors are included? Response The main sources of errors are; laser sampling error, zero level offsets, ILS error, smoothing error, atmospheric apriori temperature, atmospheric apriori pressure, surface pressure, and random noise. The total error is then computed from the sum of each error components.

- Can you specify 'regular cell measurements'? Response Regular cell measurements

are conducted one time approximately in every month.

Technical comments: - p.1 + 12: were generally agreed => generally agreed ....,both instruments generally agreed in capturing seasonal variations of the target species.... - p.2: space born => space borne ...a number of instruments deployed in various platforms (e.g., ground-based, space-borne)... - p.3: area is; => area is: .....climatic condition of the area is: the minimum temperature..... - p.4: with oil-free => with oil-free pump ....FTS is kept at 0.1 to 0.2 hPa with oil-free pump to maintain the stability of the system.... - p.5: beamspliters =< beamsplitters In Table 1. Beamsplitters

- p. 7: to these derived => to those derived (?) ...mole fractions were used only to those derived (?) below the solar zenith angle... - p. 9: - orbit, devoted => orbit. It is devoted launched on July 2, 2014 into low-Earth orbit. It is devoted to observing - can available => is available ....instrument is available in different papers...... - p.10: are varied => varied ...column amounts varied .... - p.11: - over the land => over land ...the OCO-2 data over land within.... - square => squares ..RMSE - Root Mean Squares Error....

- p.12: - and this suggesting => suggesting ....OCO-2, suggesting that the variability..... - new page within Table 4 p.13: - the source and sink of them. => their sources and sinks. ...for investigating their sources and sinks...... - outcome this => outcome of this Therefore, the outcome of this study reflects..... - Is ': : : to be withered that turns out to be weak photosynthesis : : : a grammatically correct sentence? ..... weak photosynthesis phenomenon is occurred because of low plant flourishing and CO2 reaches the highest values.......... Interactive comment on Atmos. Meas. Tech. Discuss., doi:10.5194/amt-2017-88, 2017.

Please also note the supplement to this comment: https://www.atmos-meas-tech-discuss.net/amt-2017-88/amt-2017-88-AC1-supplement.zip

Network
Camera

Lightning
rod

Rain
sensor

Solar
Tracker

GPS

Solar Beam

Dome

Detector
Compartment

IFS 125 PC   Dome PC   Camera PC

Scroll
Pump

Solar
tracker
Controller

Red Filter

HCL
Cell(10cm)

Interferometer
Compartment

Scanner
Compartment

Humidifier

Aperture

Source
Compartment

Barometer

Temperature
Controller

DATA
Server

UPS

Bruker IFS 125HR

OASIS

Ground

Dome
Controller

**Fig. 1.** Photographs of the automated FTS laboratory.

**a)**

Solar Beam

Solar
Tracker

Detector
Compartment

Si

Red Filter

HCL
Cell(10cm)

InGaAs

Interferometer
Compartment

Scanner
Compartment

Aperture

Source
Compartment

**Bruker IFS 125HR**

OASIS

**b)**

OASIS

OASIS
System

Aperture

Sun light
controller

IFS 125HR Source
compartment out sid

OASIS

Burker IFS 125HR

**Fig. 2.** Schematic views of the OASIS system.

[Figure]

**Fig. 3.** Time series of Xair, CO2, O2, surface pressure from the g-b FTS

[Figure]

**Fig. 4.** reference fig 1.

Fig. 5. reference fig 2.

[Figure]

---

## Referee Comment (RC2) · Anonymous Referee #2 · 14 Aug 2017

The manuscript "Characteristics of the Greenhouse Gas Concentration Derived from the Ground-based FTS Spectra at Anmyeondo, Korea" by Oh et al. describes the setup and first results of the only Korean observation site in the Total Carbon Column Observing Network (TCCON) which has been fully operational since 2014. Compared to other TCCN stations, the main technical difference is the addition of the OASIS system which compensates solar intensity fluctuations during the measurement.

Unfortunately, I am not happy with the manuscript in its present form. Even for a site description paper, it lacks depth. Technical details about the unique OASIS system and its performance are not provided. The description of standard TCCON procedures is

repetititve and could be left out for most parts. And the analysis of the existing time series is kept very simple with little interpretation. More sophisticated tools like model or other TCCON data comparisons are not used at all. Especially, I do not understand why the other species like CH4, CO, N2O measured by the same instrument are hardly mentioned and not used in the analysis.

In its current state, the manuscript remains a technical description of the Anmyeondo site but lacks important details and analyses about its most important technical innovation: the OASIS system. This could easily be improved and should be. For the analysis of the actual observation data, there is a a lot more than can be done. The authors could use the Darwin site paper (Deutscher et al., 2010) as an example. For a good site paper, it would be important to put the results of the site in context with the rest of the TCCON network. If the authors want to stick to a technical description of the site only, the paper might be better suited for the Copernicus Journal Geoscientific Instrumentation, Methods and Data Systems (GI).

Suggested revisions:

Section 2.2:

- what exactly does "operation is semi-automated" mean and how does it affect e.g. the number of measurements as opposed to fully autmated like many other TCCON instruments?

- the many details in the FTS and solar tracker description are only useful for TCCON experts. For the general reader, you should mention that the part described here is a standard setup for a TCCON site (except for OASIS).

Section 2.3: OASIS

I think this is the most interesting technical part of this TCCON site but the description and analysis is not thorough enough. Especially, I miss a detailed discussion of the pros and cons of a system like OASIS. For example:

[Figure]

- what is the dynamic range of OASIS?

- how large is the quality difference of clear-sky spectra with OASIS compared to without?

- how does this spectral quality difference translate into retrieval quality inprovement?

- in the example in Fig. 5, why is the signal with OASIS lower than without? Does the better stability with OASIS compensate the loss in intensity and hence in signal-to-noise?

- do you log what OASIS is doing? Can you still distinguish between observations with truly clear sky and such with thin clouds or other intensity fluctuations? Maybe some would better be dropped rather than compensated.

- is a system like OASIS worth the effort? How many more spectra do you get compared to the TCCON approach of dropping ones with SIV>5%? In Fig. 5, it looks like there were only a few events with strons drops in intensity. And I wonder if they could actually be corrected by OASIS. Certainly, a thick cloud moving in from of the sun cannot be compensated.

- how does OASIS affect the pointing accuracy of your solar tracker?

Section 2.4:

This whole subsection only describes standard TCCON retrieval procedure without any obvious site-specific adaptions. I think the whole subsection can be left out and be replaced by a single sentence and a reference to Wunch et al. 2015.

Section 3.1:

- you should explain a little better what Xair is and how it can be used as an indicator of stability.

- what are the plots in Fig. 8 showing? Obviously not single retrievals! Are these daily

means or medians or something else?

- Plotting and discussing CO2 and O2 separately here is a poor choice. The whole idea behing TCCON is to remove airmass-related effects to produce high-precision observations. What we see here is simply the change in airmass probably due to the seasonal change of the sun's position in the sky (and a small part from ground pressure changes). All carbon-cycle related effects are completely hidden by this effect.

Section 3.2:

- I don't aunderstand the argument about the comparison between between g-b FTS and OCO-2. A priori profiles and averaging kernels are available for both observations. What CO2 profile do you mean with "... since we do not have the CO2 profile that reflects the actual variability over the measurement site."?

Section 3.3:

- Is this really all you can see: XCO2 variability because of photosynthesis? Are there no in-situ observations nearby so one could separate CO2 in the Planetary Boundary Layer (PBL) from CO2 in the free troposphere or at least look for differences in PBL and total column?

- Especially in this section, the other observed species like CH4, CO, and N2O might have been really useful. I doubt that all you can see at your site are local effects and maybe some seasonal background variation. There must be transport from other regions which would probably show up in CH4 or CO.

References:

- the authors should include their own TCCON data citation reference and DOI in their reference list!

Minor corrections:

- please make sure that all acronyms and abbreviations (liek "g-b") are defined in the

main text, even if they have been defined in the abstract already.

- p. 1, l. 31: "G-b FTS" is not a very good choive for a keyword. Neither is "OASIS" as it is not a unique term and also not a well-established acronym (yet).

- p. 2, l. 15: "TCCON achieves the accuracy and precision in measuring the total column of $CO_2$ ..." -> TCON achieves this precision and accuracy for the column-averaged dry air mole fraction of $CO_2$ ($XCO_2$), not for the total column!

- p. 2, l. 12: you mention "several atmospheric GHGs" but you neither say which nor discuss them in any way in this manuscript. Why?

- p. 2, l. 25: "a new home made OASIS system" sounds as if "OASIS" was an established acronym. It is not, it is just your internal name for your device. It might also be better to define the acronym OASIS in the main text rather than in the abstract. My suggestion for the sentence would be "One of the interesting issues in this work is a new home made addition to our g-b FTS instrument (see Sect. 2.3) that reduces the solar intensity variations from the 5% maximum allowed in TCCON to less than 2%."

- p. 2, l. 25: "SIV" instead of "SVI"! In fact, you don't need this acronym at all. It is TCCON jargon and only used three times in the whole manuscript (and you spelled it out each time!).

- p. 2, l. 27-28: there is no need to provide an outline of the sections and numbers. Scientific papers typically don't have a table of contents. Just drop these two lines.

- p. 3, l- 3: Please replace "G-b" with "g-b" throughout the text unless it starts a sentence.

- p. 3, l. 5: "... Seoul, the capital city of Korea." -> I don't want get into politics here but isn't the country officially named "Republic of Korea"? "South Korea" would probably also be clear, maybe even clearer to the general reader.

- p. 3, l. 22-23: avoid the line break for "A 547". In fact, I believe the tracker model

number is "A547" (w/o space).

- p. 3, l. 23: "... are about 0° to 315° and 10° to 85° degrees ..." -> (1) "°" and "degrees" are redundant, (2) is the elevation range really only 10 to 85 degrees? Does that mean the tracker cannot point to the horizon or zenith at all?

- p. 4, l. 3: "oil-free" -> "vacuum pump" is missing. Is the pump running continuously?

- p. 5, l. 9: "... voltage ranges of approximately 0 to 219 mV." This information is hardly useful for anyone outside your department. Especially since you claim that "... the detail characteristic of the operation is beyond the scope of this paper."

- p. 5, l. 13-14: "the intensity of the incoming light occurred due to changes in thin clouds and aerosols loads or interceptions by any other objects along the line of sight over the measurement site." -> thin clouds is clear but aerosol load should not really change during a 2-minute measurement. And what objects could be passing the line of sight often enough to justify such a system?

- p. 6, l. 13: GGG is not developed "by JPL" even though the main developer works there. But there are also other main developers who work at different institutions even ouside Caltec/JPL. It would be correvt to say that GGG is developed by the TCCON community.

- p. 8, l. 9: "LINFIT" -> "LINEFIT"!

Figures:

- Fig. 1: (1) picture quality is not very good, (2) country borders and maybe the location of Seould would be helpful, (3) For the insets: the upper one is clear but what is the lower one? The labels are Korean only.

- Fig. 2: (1) "server" instead of "sever", (2) you used "solar tracker" throughout the text, so you should not use "sun tracker" in the figure, (3) "Photographs of the automated FTS laboratory."

- Fig. 3: how is signal-to-noise defined here?

- Fig. 4: low quality/resolution

- Fig. 5: (1) low quality/resolution, (2) is this the same day for both plots? (3) why does signal drop off to the right with OASIS even though start time is earlier? (3) better plot this over solar zenith angle than over time!

- Fig. 6: very low quality with obvious JPEG compression artifacts. This should be redone in a lossless compression format like PNG or a vector format like PDF!

- Fig. 7: (1) not referenced in the text at all! (2) Should probably belong to Sec. 2.4 which means it should appear before (!) Fig. 6. (3) I don't know why this Figure is even part of the manuscript. Is this an original figure created by the authors or taken from somewhere else? (4) similar quality problem as with the other figures. The box labels are basically unreadable.

- Fig. 8: (1) low quality/resolution (2) why not just plot XCO2? The variations in column are mostly due to seasonl airmass variation (as can be seen in O2 column). XCO2 would tell you something about carbon-cycle related effects at your site!

- Fig. 9: unlike the other figures, this one has acceptable quality. I would still suggest to plot daily medians instead of means.

---

## Author Comment (AC2) · 2 Oct 2017

General comments: First of all, we would like to strongly appreciate referee's very constructive and valuable comments, suggestions, and feedbacks on the manuscript. On the basis of this, we have tried to address all the issues raised on this manuscript. We have discussed the feature of OASIS system and its performance in a detail manner in the revised manuscript. We have included other TCCON site data for comparison purpose, and also added some species such as CO, and CH4 derived from Anmyeondo FTS instrument. Comments: Section 2.2: What exactly does "operation is semi-automated" mean and how does it affect e.g. the number of measurements as

opposed to fully automated like many other TCCON instruments?

Response: The FTS instrument is operated in semi-automated, which mean that some systems are operated by manual (someone should be there for controlling certain systems). However, this mode of operation does not affect the measurements.

Section 2.3: OASIS I think this is the most interesting technical part of this TCCON site but the description and analysis is not thorough enough. Especially, I miss a detailed discussion of the pros and cons of a system like OASIS. For example: what is the dynamic range of OASIS?

Response: We tried to elaborate regarding Operational Automatic System for the Intensity of Sunray (OASIS) system to some extent in section 2.3, which may not be sufficient to get in-depth information about it. We planned to address all issues like the pros and cons of this system on the measurements, as well as other technical details based on sufficient experiments on a special issue. (*The OASIS part was decided to write a separate paper on TCCON.)

How large is the quality difference of clear-sky spectra with OASIS compared to without? - In the example in Fig. 5, why is the signal with OASIS lower than without?

Response: In this particular example, the spectra were taken with and without OASIS April 04, 2015 (starting time in the case of without OASIS was 06:12:03 and ending time 08:46:40 while the starting time in the case of OASIS was 04:31:00 and approximately ending time 05:40:00). The solar intensity differences are occurred due to measurement time differences. Unfortunately, we did not conduct experiment in assessment of the quality of spectra measured with and without OASIS during clear sky condition. In next work, this will be examined. (*The OASIS part was decided to write a separate paper on TCCON.)

Does the better stability with OASIS compensate the loss in intensity and hence in signal-to noise?

Response: Yes, it would improve well the stability and signal-to-noise ratio of the spectra. (*The OASIS part was decided to write a separate paper on TCCON.)

do you log what OASIS is doing? Can you still distinguish between observations with truly clear sky and such with thin clouds or other intensity fluctuations? May be some would better be dropped rather than compensated. Is a system like OASIS worth the effort? How many more spectra do you get compared to the TCCON approach of dropping ones with SIV>5%?

Response: Yes, this OASIS system controls the aperture size based on the external sun light intensity. In the meantime, we do not have clear idea to distinguish observations with truly clear sky and such with thin clouds since we did not perform experiments. We obtained around 1230 number of spectra more as compared to TCCON approach of dropping ones with SIV > 5%. It is required further effort to briefly explain the impact of the OASIS system on the measurements. (*The OASIS part was decided to write a separate paper on TCCON.)

In Fig. 5, it looks like there were only a few events with strong drops in intensity. And I wonder if they could actually be corrected by OASIS. Certainly, a thick cloud moving in from of the sun cannot be compensated. How does OASIS affect the pointing accuracy of your solar tracker?

Response: Yes surely, a thick cloud moving in from of the sun cannot be compensated. (*The OASIS part was decided to write a separate paper on TCCON.)

Section 2.4: This whole subsection only describes standard TCCON retrieval procedure without any obvious site-specific adaptions. I think the whole subsection can be left out and be replaced by a single sentence and a reference to Wunch et al. 2015.

Response: We have removed the unnecessary part of Section 2.4 retrieval methodology, and modified this section; please see section 2.5 Data processing in revised manuscript. Section 3.1: - you should explain a little better what Xair is and how it can

be used as an indicator of stability. what are the plots in Fig. 8 showing? Obviously not single retrievals! Are these daily means or medians or something else?

Response: The Xair would be unity for an ideal retrieval, however, due to spectroscopic limitations there is a TCCON wide bias and solar zenith angle (SZA) dependence. The Xair is a useful indicator of the quality of measurements, with retrievals deviating more than 1% from the nominal value of 0.98 demonstrating systematic error. Initially, Fig. 8 showed the time series of Xair, surface pressure, and column amounts of O2 and CO2 in daily means in the period between February 2014 and December 2015, but we have re-plotted this figure again, where we considered only Xair and others are excluded.

- Plotting and discussing CO2 and O2 separately here is a poor choice. The whole idea behind TCCON is to remove airmass-related effects to produce high-precision observations. What we see here is simply the change in airmass probably due to the seasonal change of the sun's position in the sky (and a small part from ground pressure changes). All carbon-cycle related effects are completely hidden by this effect.

Response: We appreciate this very nice comment. We understood that discussing CO2 and O2 separately is not relevant so that we removed this discussion part.

Section 3.2: - I don't understand the argument about the comparison between g-b FTS and OCO-2. A priori profiles and averaging kernels are available for both observations. What CO2 profile do you mean with "... since we do not have the CO2 profile that reflects the actual variability over the measurement site."?

Response: We have improved the arguments about the comparison between g-b FTS and OCO-2. Please see the discussion in section 3.4 in revised manuscript.

Section 3.3: - Is this really all you can see: XCO2 variability because of photosynthesis? Are there no in-situ observations nearby so one could separate CO2 in the Planetary Boundary Layer (PBL) from CO2 in the free troposphere or at least look for differences in PBL and total column?

Response: We strongly appreciate the comment. We included the in-situ tower observation data and compared the seasonal cycle of $CO_2$ with g-b FTS $XCO_2$. The seasonal cycle of FTS $XCO_2$ followed nearly same pattern as that of in-situ observations, this would suggest that seasonal cycle of $CO_2$ is most likely controlled by the imbalance of terrestrial ecosystem exchange, even though it is required further work to examine other effect like the role of transport. Please see section 3.1 in revised manuscript.

- Especially in this section, the other observed species like $CH_4$, $CO$, and $N_2O$ might have been really useful. I doubt that all you can see at your site are local effects and maybe some seasonal background variation. There must be transport from other regions which would probably show up in $CH_4$ or $CO$.

Response: In addition of $XCO_2$, we have also considered other species such as $XCO$ and $XCH_4$. The $XCO_2$ along with the retrievals of $XCO$ and $XCH_4$ obtained from g-b FTS spectra are presented in Figure 8 (panel a-c), in the time period of February 2014–December 2016. (Please see section 3.1). Furthermore, we have discussed the relation between $XCO$ and $XCO_2$ at our site, which is presented in section 3.2 in revised manuscript.

Minor corrections: - please make sure that all acronyms and abbreviations (liek "g-b") are defined in the main text, even if they have been defined in the abstract already.

Response: We corrected and checked all acronyms and abbreviations throughout the text in the manuscript.

- p. 1, l. 31: "G-b FTS" is not a very good choice for a keyword. Neither is "OASIS" as it is not a unique term and also not a well-established acronym (yet).

Response: We have removed "G-b FTS" and "OASIS" as key words. (*The OASIS part was decided to write a separate paper on TCCON.)

- p. 2, l. 15: "TCCON achieves the accuracy and precision in measuring the total col-

umn of CO2 ..." -> TCON achieves this precision and accuracy for the column averaged dry air mole fraction of CO2 (XCO2), not for the total column!

Response: We have corrected as "TCCON achieves the accuracy and precision in measuring the column averaged dry air mole fraction of CO2 about 0.25 %..........."

- p. 2, l. 12: you mention "several atmospheric GHGs" but you neither say which nor discuss them in any way in this manuscript. Why?

Response: It is very nice comment. We described the GHGs that have been measured with our instruments at Anmyeondo site. Those prominent GHGs are CO2, CH4, CO, N2O, and H2O. We included CH4 and CO results obtained from g-b FTS and discussed.

- p. 2, l. 25: "a new home made OASIS system" sounds as if "OASIS" was an established acronym. It is not, it is just your internal name for your device. It might also be better to define the acronym OASIS in the main text rather than in the abstract. My suggestion for the sentence would be "One of the interesting issues in this work is a new home made addition to our g-b FTS instrument (see Sect. 2.3) that reduces the solar intensity variations from the 5% maximum allowed in TCCON to less than 2%."

Response: Thanks to the comment. We have defined the acronym OASIS in the main text as well. We have replaced the previous sentence written in Sect. 2.3 by "One of the interesting issues in this work is a new home made addition to our g-b FTS instrument (see Sect. 2.3) that reduces the solar intensity variations from the 5% maximum allowed in TCCON to less than 2%." (*The OASIS part was decided to write a separate paper on TCCON.)

- p. 2, l. 25: "SIV" instead of "SVI"! In fact, you don't need this acronym at all. It is TCCON jargon and only used three times in the whole manuscript (and you spelled it out each time!).

Response: We made correction "SVI" by "SIV".

- p. 2, l. 27-28: there is no need to provide an outline of the sections and numbers. Scientific papers typically don't have a table of contents. Just drop these two lines.

Response: We improved it. Please see the last paragraph of the introduction section in revised manuscript.

- p. 3, l- 3: Please replace "G-b" with "g-b" throughout the text unless it starts a sentence.

Response: We corrected it.

- p. 3, l. 5: "... Seoul, the capital city of Korea." -> I don't want get into politics here but isn't the country officially named "Republic of Korea"? "South Korea" would probably also be clear, maybe even clearer to the general reader.

Response: Thanks for the comment. The country official name is "Republic of Korea". We corrected the sentence "... Seoul, the capital city of Republic of Korea."

- p. 3, l. 22-23: avoid the line break for "A 547". In fact, I believe the tracker model number is "A547" (w/o space).

Response: Replaced "A 547" by "A547".

- p. 3, l. 23: "... are about 0_ to 315_ and 10_ to 85_ degrees ..." -> (1) "_" and "degrees" are redundant, (2) is the elevation range really only 10 to 85 degrees? Does that mean the tracker cannot point to the horizon or zenith at all?

Response: We removed the redundant of "_" and "degrees" from the text. The tracker can point to the horizon and zenith as well.

- p. 4, l. 3: "oil-free" -> "vacuum pump" is missing. Is the pump running continuously?

Response: The vacuum pump is running continuously.

- p. 5, l. 9: "... voltage ranges of approximately 0 to 219 mV." This information is hardly useful for anyone outside your department. Especially since you claim that "... the

detail characteristic of the operation is beyond the scope of this paper."

Response: We omitted "...voltage ranges of approximately 0 to 219 mV...." from the text, since the detail characteristic of the operation is beyond the scope of this paper.

- p. 5, l. 13-14: "the intensity of the incoming light occurred due to changes in thin clouds and aerosols loads or interceptions by any other objects along the line of sight over the measurement site." -> thin clouds is clear but aerosol load should not really change during a 2-minute measurement. And what objects could be passing the line of sight often enough to justify such a system?

Response: We understood that this sentence "...interceptions by any other objects along the line of...." is irrelevant so that we omitted the sentence "...interceptions by any other objects ...." in the text.

- p. 6, l. 13: GGG is not developed "by JPL" even though the main developer works there. But there are also other main developers who work at different institutions even outside Caltec/JPL. It would be correct to say that GGG is developed by the TCCON community.

Response: We thank the referee's comments. We corrected it accordingly.

- p. 8, l. 9: "LINFIT" -> "LINEFIT"! Response: Corrected "LINFIT" by "LINEFIT"

Figures Response: All figures have been replaced. Data analysis period and quality improved.

- Fig. 1: (1) picture quality is not very good, (2) country borders and maybe the location of Seoul would be helpful, (3) For the insets: the upper one is clear but what is the lower one? The labels are Korean only.

- Fig. 2: (1) "server" instead of "sever", (2) you used "solar tracker" throughout the text, so you should not use "sun tracker" in the figure, (3) "Photographs of the automated FTS laboratory."

- Fig. 3: how is signal-to-noise defined here?

- Fig. 4: low quality/resolution

- Fig. 5: (1) low quality/resolution, (2) is this the same day for both plots? (3) why does signal drop off to the right with OASIS even though start time is earlier? (3) better plot this over solar zenith angle than over time!

- Fig. 6: very low quality with obvious JPEG compression artifacts. This should be redone in a lossless compression format like PNG or a vector format like PDF!

- Fig. 7: (1) not referenced in the text at all! (2) Should probably belong to Sec. 2.4 which means it should appear before (!) Fig. 6. (3) I don't know why this Figure is even part of the manuscript. Is this an original figure created by the authors or taken from somewhere else? (4) similar quality problem as with the other figures. The box labels are basically unreadable.

- Fig. 8: (1) low quality/resolution (2) why not just plot XCO2? The variations in column are mostly due to seasonal airmass variation (as can be seen in O2 column). XCO2 would tell you something about carbon-cycle related effects at your site!

- Fig. 9: unlike the other figures, this one has acceptable quality. I would still suggest to plot daily medians instead of means.

Please also note the supplement to this comment:
https://www.atmos-meas-tech-discuss.net/amt-2017-88/amt-2017-88-AC2-supplement.pdf

———————————————————

Fig. 1. Anmyeodo (AMY) g-b FTS station

**Fig. 2.** Photographs of the automated FTS laboratory

**Fig. 3.** Single spectrum

- ● IMECC
- ■ AIRCORE
- ◆ BEECHCRAFT
- ▲ COBRA
- ▼ HIPPO
- ▶ INTEX
- ◀ INTEX-COBRA
- ◆ KORUS
- ✳ LEARJET
- ★ START08
- ✳ TWP-ICE
- - - y=0.9898+/-0.0005x; R²=0.995

Site labels:
Anmyeondo
**Bialystok**
Bremen
Darwin
**Four Corners**
Garmisch
Jena
Karlsruhe
Lauder
Lamont
Orleans
**Park Falls**
Tsukuba
Wollongong

Axis labels:
TCCON XCO2 (ppm) (y-axis)
Aircraft XCO2 (ppm) (x-axis)

**Fig. 4.** The result of comparison between aircraft observation and TCCON concentration

**Fig. 5.** The result of comparison between aircraft observation and TCCON concentration

**Fig. 6.** Phase error

**Fig. 7.** Modulation efficiency

[Figure]

**Fig. 8.** Time series of LSE (top panel) and Xair (bottom panel) from the g-b FTS during 2014-2016

[Figure]

**Fig. 9.** Time series of XCO2, XCO, and XCH4 from top to bottom panels (a-c), respectively in the period between February 2014 to December 2016.

[Figure]

**Fig. 10.** Left panel shows the time series of FTS XCO2 and in-situ tower CO2 on monthly mean basis, whereas right panel depicts annual cycle

[Figure]

**Fig. 11.** Correlation between XCO2 versus XCO anomalies at Anmyeondo FTS site between February 2014 to December 2016, excluding summer data

[Figure]

**Fig. 12.** Time series of XCO2 retrieval from Anmyeondo FTS and Saga FTS in the period of February 2014 to December 2016 .

[Figure]

**Fig. 13.** Left panel: The time series of XCO2 from the g-b FTS (blue triangle) and OCO-2 (red triangle) over the Anmyeondo station from February 2014 to December 2016. Right panel: The linear regression curve b

---

## Author Response (AR1)

T. Blumenstock (Referee)

thomas.blumenstock@kit.edu

General comments:

The authors present a new TCCON site in Korea. This paper characterizes the instrumentation and gives an example of its application: inter-comparison with OCO-2 satellite data. This site really fills a gap in the existing TCCON network and will be very useful to assessing sink and sources of GHGs. The data and also the comparison with OCO-2 data are of good quality. The subject is appropriate for publication in AMT. The paper is well written and I recommend publication after major revisions, in particular a more comprehensive description.

**First of all, we would like to strongly appreciate referee's very constructive and valuable comments on the manuscript. We have tried to address all the issues (major and minor comments) raised on this paper one by one. The referee makes strong comments to give more details about OASIS and its influence on ILS (Instrumental Line Shape) by including some illustrations so that we added some more brief on it. Our replies and respective changes are described below. Technical comments regarding spellings and grammatical errors are corrected in the final version of the manuscript.**

**Major comment:**

A specific feature of the described instrumentation is the so called OASIS (Operational Automatic System for Intensity of Sunray) system. While analog systems are used in active remote sensing systems, for example laser output control in LIDAR systems, an intensity control of passive systems is typically not used. In the TCCON network the variability of the DC signal is used to quality check and correct the recorded interferograms and resulting spectra. Since you remove this signal you cannot apply this kind of quality check anymore. Do you record and use the actual setting of the aperture to do so?

**Response**

**In addition to OASIS system, we have also simultaneously used the variability of DC signal similar to TCCON network for quality control of the spectra and its retrieval results. Yes, the spectra are recorded with the actual setting of the aperture having a diameter of 0.8 mm throughout the observation period.**

If the motivation to introduce such a system is to limit the intensity to avoid non-linear response a smaller constant aperture or smaller preamp gain or a smaller sensitivity of the detector might be more appropriate. Would you please add a statement for the motivation to add this system? Or a comparison of $XCO_2$ time series recorded with and without OASIS system which might demonstrate the difference, for example in terms of signal to noise ratio.

**Response**

**The OASIS system is developed for improving the quality of the spectra. To ensure the quality of the spectra, this system will be useful for minimizing the noise that induced in the spectra due to rapid intensity fluctuations of the incoming solar radiation that reaches to the instrument. This rapid intensity fluctuations are be occurred in the presence of clouds, aerosol loading etc. along the path of incoming radiation within the instrument field of view. To minimize this intensity fluctuations due to the changing weather conditions, OASIS system regulates in such a way that by varying the aperture size at the source compartment based on the signals from photon sensor which depends on the levels of incoming sunlight intensity. Thereby, it avoids non-linear response of a smaller constant aperture or smaller preamp gain or a smaller sensitivity of the detector. In this study, we are not able to show the whole time series of $XCO_2$ without OASIS system during the study period since all spectra that are used for analysis of species are obtained after the OASIS system equipped to our FTS spectrometer. However, for a typical example, we illustrated the time series of $XCO_2$ in both cases.**

**Based on TCCON community suggestions regarding the OASIS system, it would be recommended to use a consistent g-b FTS measurement set up throughout TCCON network so that we plan to fix a constant aperture size at the source compartment during the FTS operation at Anmyeondo station.**

Where is the variable aperture positioned?

**Response**

**A variable aperture is placed inside the OASIS system which is at the source compartment.**

Is it in the parallel or focused beam?

**Response**

**It is a focused beam.**

Did you check the influence on the ILS (Instrumental Line Shape) due to the variable aperture while scanning?

**Response**

**Yes, we assessed the influence of ILS due to the variable aperture and the result showed that it has no impact on the ILS.**

I assume a lamp was used and hence the OASIS system was not active while performing cell measurements. Cell measurements using the sun as source might be an option to check the ILS while the OASIS system is active. Or, if the HCl lines in the atmospheric spectrum are covered by interfering species you might do cell measurements with the lamp using different fixed aperture settings to check the influence of the OASIS system on the ILS. How does your system and its influence on the ILS compares with the results of the recent paper by Sun et al, AMT, 2017 on the 'Sensitivity of instrumental line shape monitoring for the ground-based high-resolution FTIR spectrometer with respect to different optical attenuators'? While most of the site complies with the TCCON standard setup the OASIS system does not. Therefore a more detailed description is needed as well as a discussion on its influence on the ILS.

**Response**

**We have carried out experiments to investigate the influences of ILS due to the presence of OASIS system, and then considered HCl cell measurements using sun as source while OASIS system active and tungsten lamp as a source while OASIS inactive. The result confirmed that the ILS was not affected by the variable aperture during the operation of OASIS system. Sun et al. (2017) reported the detailed characteristics of the ILS with respect to applications of different optical attenuators to FTIR spectrometers within the TCCON and NDACC networks. They used both lamp and sun cell measurements which were conducted after the insertion of five different attenuators in front of and behind the interferometer. In Sun et al. (2017) paper, the ILS result was indicated by considering optical attenuator no .1 which is in good agreement with our findings.**

**Specific comments:**

- In Chapter 3.1 the time series of the $O_2$ columns is compared with atmospheric pressure. Therefore including surface pressure in Fig. 8 might support your statement.

**Response**

**The time series of surface pressure is included in bottom panel of Fig. 8 and compared with the time series of $O_2$ column.**

- The errors are shown in Fig. 9. How is the error calculated and which sources of errors are included?

**Response**

**The main sources of errors are; laser sampling error, zero level offsets, ILS error, smoothing error, atmospheric apriori temperature, atmospheric apriori pressure, surface**

**pressure, and random noise. The total error is then computed from the sum of each error components.**

- Can you specify 'regular cell measurements'?

**Response**

**Regular cell measurements are conducted one time approximately in every month.**

**Technical comments:**

- p.1 + 12: were generally agreed => generally agreed

….,both instruments generally agreed in capturing seasonal variations of the target species….

- p.2: space born => space borne

…a number of instruments deployed in various platforms (e.g., ground-based, space-borne)…

- p.3: area is; => area is:

…..climatic condition of the area is: the minimum temperature…..

- p.4: with oil-free => with oil-free pump

….FTS is kept at 0.1 to 0.2 hPa with oil-free pump to maintain the stability of the system….

- p.5: beamspliters =< beamsplitters

In Table 1. Beamsplitters

- p. 7: to these derived => to those derived (?)

…mole fractions were used only to those derived (?) below the solar zenith angle…

- p. 9:

- orbit, devoted => orbit. It is devoted

launched on July 2, 2014 into low-Earth orbit. It is devoted to observing

- can available => is available

….instrument is available in different papers……

- p.10: are varied => varied

…column amounts varied ….

- p.11:

- over the land => over land

…the OCO-2 data over land within….

- square => squares

..RMSE - Root Mean Squares Error….

- p.12:

- and this suggesting => suggesting

….OCO-2, suggesting that the variability.....

- new page within Table 4

p.13:

- the source and sink of them. => their sources and sinks.

...for investigating their sources and sinks......

- outcome this => outcome of this

Therefore, the outcome of this study reflects.....

- Is ': : : to be withered that turns out to be weak photosynthesis : : : a grammatically correct sentence?

….. weak photosynthesis phenomenon is occurred because of low plant flourishing and $CO_2$ reaches the highest values..........

**First referee revision.**

[revised manuscript text omitted]

| Light Source | Tungsten | Solar(Sun) | Solar(Sun) | Range |
|---|---|---|---|---|
| S/N (signal to noise ratio) | 183.2 : 1 | 162.7 : 1 | 167.1 : 1 | _ |
| Resolution (5687.65cm$^{-1}$) | 0.0137 | 0.0143 | 0.0145 | 0.0135~0.0149 |
| Transmission | -0.0005 to 0.0005 | -0.001 to 0.001 | -0.001 to 0.001 | _ |
| Mod. eff | 99.99 % | 99.98 % | 99.96 % | 99.96 ~99.99 % |
| OASIS run | **OFF** | **OFF** | **ON** | |
| Analysis Parameter | (Same Parameter) Resolution: 0.015cm-1, Scans: 50, Beamspliter: CaF2, Aperture: 0.8mm, Detector: RT-InGaAs DC, Scanner velocity: 10kHz, High pass filter: open, Low pass filter: 10kHz (Different Parameter) Source setting: Emission back parallel input/ NIR | | | |

[Figure]

**Figure 4:** Schematic views of the OASIS system

[revised manuscript text omitted]

**General comments:**

First of all, we would like to strongly appreciate referee's very constructive and valuable comments, suggestions, and feedbacks on the manuscript. On the basis of this, we have tried to address all the issues raised on this manuscript. We have discussed the feature of OASIS system and its performance in a detail manner in the revised manuscript. We have included other TCCON site data for comparison purpose, and also added some species such as CO, and $CH_4$ derived from Anmyeondo FTS instrument.

**Comments:**

Section 2.2: What exactly does "operation is semi-automated" mean and how does it affect e.g. the number of measurements as opposed to fully automated like many other TCCON instruments?

Response: The FTS instrument is operated in semi-automated, which mean that some systems are operated by manual (someone should be there for controlling certain systems). However, this mode of operation does not affect the measurements.

Section 2.3: OASIS

I think this is the most interesting technical part of this TCCON site but the description and analysis is not thorough enough. Especially, I miss a detailed discussion of the pros and cons of a system like OASIS. For example: what is the dynamic range of OASIS?

Response: We tried to elaborate regarding Operational Automatic System for the Intensity of Sunray (OASIS) system to some extent in section 2.3, which may not be sufficient to get in-depth information about it. We planned to address all issues like the pros and cons of this system on the measurements, as well as other technical details based on sufficient experiments on a special issue.
(*The OASIS part was decided to write a separate paper on TCCON.)

How large is the quality difference of clear-sky spectra with OASIS compared to without? - In the example in Fig. 5, why is the signal with OASIS lower than without?

Response: In this particular example, the spectra were taken with and without OASIS April 04, 2015 (starting time in the case of without OASIS was 06:12:03 and ending time 08:46:40 while the starting time in the case of OASIS was 04:31:00 and approximately ending time 05:40:00). The solar intensity differences are occurred due to measurement time differences. Unfortunately, we did not conduct experiment in assessment of the quality of spectra measured with and without OASIS during clear sky condition. In next work, this will be examined. (*The OASIS part was decided to write a separate paper on TCCON.)

Does the better stability with OASIS compensate the loss in intensity and hence in signal-to-noise?

Response: Yes, it would improve well the stability and signal-to-noise ratio of the spectra. (*The OASIS part was decided to write a separate paper on TCCON.)

do you log what OASIS is doing? Can you still distinguish between observations with truly clear sky and such with thin clouds or other intensity fluctuations? May be some would better be dropped rather than compensated. Is a system like OASIS worth the effort? How many more spectra do you get compared to the TCCON approach of dropping ones with SIV>5%?

Response: Yes, this OASIS system controls the aperture size based on the external sun light intensity. In the meantime, we do not have clear idea to distinguish observations with truly clear sky and such with thin clouds since we did not perform experiments. We obtained around 1230 number of spectra more as compared to TCCON approach of dropping ones with SIV > 5%. It is required further effort to briefly explain the impact of the OASIS system on the measurements. (*The OASIS part was decided to write a separate paper on TCCON.)

In Fig. 5, it looks like there were only a few events with strong drops in intensity. And I wonder if they could actually be corrected by OASIS. Certainly, a thick cloud moving in from of the sun cannot be compensated. How does OASIS affect the pointing accuracy of your solar tracker?

Response: Yes surely, a thick cloud moving in from of the sun cannot be compensated. (*The OASIS part was decided to write a separate paper on TCCON.)

Section 2.4:
This whole subsection only describes standard TCCON retrieval procedure without any obvious site-specific adaptions. I think the whole subsection can be left out and be replaced by a single sentence and a reference to Wunch et al. 2015.

Response: We have removed the unnecessary part of Section 2.4 retrieval methodology, and modified this section; please see section 2.5 Data processing in revised manuscript.

Section 3.1:
- you should explain a little better what $X_{air}$ is and how it can be used as an indicator of stability. what are the plots in Fig. 8 showing? Obviously not single retrievals! Are these daily means or medians or something else?

Response: The $X_{air}$ would be unity for an ideal retrieval, however, due to spectroscopic limitations there is a TCCON wide bias and solar zenith angle (SZA) dependence. The $X_{air}$ is a useful indicator of the quality of measurements, with retrievals deviating more than 1% from the nominal value of 0.98 demonstrating systematic error. Initially, Fig. 8 showed the time series of $X_{air}$, surface pressure, and column amounts of $O_2$ and $CO_2$ in daily means in the period between February 2014 and December 2015, but we have re-plotted this figure again, where we considered only $X_{air}$ and others are excluded.

- Plotting and discussing $CO_2$ and $O_2$ separately here is a poor choice. The whole idea behind TCCON is to remove airmass-related effects to produce high-precision observations. What we see here is simply the change in airmass probably due to the seasonal change of the sun's position in the sky (and a small part from ground pressure changes). All carbon-cycle related effects are completely hidden by this effect.

Response: We appreciate this very nice comment. We understood that discussing $CO_2$ and $O_2$ separately is not relevant so that we removed this discussion part.

Section 3.2:
- I don't understand the argument about the comparison between g-b FTS and OCO-2. A priori profiles and averaging kernels are available for both observations. What $CO_2$ profile do you mean with "... since we do not have the $CO_2$ profile that reflects the actual variability over the measurement site."?

Response: We have improved the arguments about the comparison between g-b FTS and OCO-2. Please see the discussion in section 3.4 in revised manuscript.

Section 3.3:
- Is this really all you can see: $XCO_2$ variability because of photosynthesis? Are there no in-situ observations nearby so one could separate $CO_2$ in the Planetary Boundary Layer (PBL) from $CO_2$ in the free troposphere or at least look for differences in PBL and total column?

Response: We strongly appreciate the comment. We included the in-situ tower observation data and compared the seasonal cycle of $CO_2$ with g-b FTS $XCO_2$. The seasonal cycle of FTS $XCO_2$ followed nearly same pattern as that of in-situ observations, this would suggest that seasonal cycle of $CO_2$ is most likely controlled by the imbalance of terrestrial ecosystem exchange, even though it is required further work to examine other effect like the role of transport. Please see section 3.1 in revised manuscript.

- Especially in this section, the other observed species like $CH_4$, $CO$, and $N_2O$ might have been really useful. I doubt that all you can see at your site are local effects and maybe some seasonal background variation. There must be transport from other regions which would probably show up in $CH_4$ or $CO$.

Response: In addition of $XCO_2$, we have also considered other species such as $XCO$ and $XCH_4$. The $XCO_2$ along with the retrievals of $XCO$ and $XCH_4$ obtained from g-b FTS spectra are

presented in Figure 8 (panel a-c), in the time period of February 2014–December 2016. (Please see section 3.1). Furthermore, we have discussed the relation between XCO and $XCO_2$ at our site, which is presented in section 3.2 in revised manuscript.

**Minor corrections:**
- please make sure that all acronyms and abbreviations (liek "g-b") are defined in the main text, even if they have been defined in the abstract already.

Response: We corrected and checked all acronyms and abbreviations throughout the text in the manuscript.

- p. 1, l. 31: "G-b FTS" is not a very good choice for a keyword. Neither is "OASIS" as it is not a unique term and also not a well-established acronym (yet).

Response: We have removed "G-b FTS" and "OASIS" as key words.
(*The OASIS part was decided to write a separate paper on TCCON.)

- p. 2, l. 15: "TCCON achieves the accuracy and precision in measuring the total column of CO2 ..." -> TCON achieves this precision and accuracy for the column averaged dry air mole fraction of CO2 ($XCO_2$), not for the total column!

Response: We have corrected as "TCCON achieves the accuracy and precision in measuring the column averaged dry air mole fraction of $CO_2$ about 0.25 %............"

- p. 2, l. 12: you mention "several atmospheric GHGs" but you neither say which nor discuss them in any way in this manuscript. Why?

Response: It is very nice comment. We described the GHGs that have been measured with our instruments at Anmyeondo site. Those prominent GHGs are $CO_2$, $CH_4$, CO, $N_2O$, and $H_2O$. We included $CH_4$ and CO results obtained from g-b FTS and discussed.

- p. 2, l. 25: "a new home made OASIS system" sounds as if "OASIS" was an established acronym. It is not, it is just your internal name for your device. It might also be better to define the acronym OASIS in the main text rather than in the abstract. My suggestion for the sentence would be "One of the interesting issues in this work is a new home made addition to our g-b FTS instrument (see Sect. 2.3) that reduces the solar intensity variations from the 5% maximum allowed in TCCON to less than 2%."

Response: Thanks to the comment. We have defined the acronym OASIS in the main text as well. We have replaced the previous sentence written in Sect. 2.3 by "One of the interesting issues in

this work is a new home made addition to our g-b FTS instrument (see Sect. 2.3) that reduces the solar intensity variations from the 5% maximum allowed in TCCON to less than 2%."
(*The OASIS part was decided to write a separate paper on TCCON.)

- p. 2, l. 25: "SIV" instead of "SVI"! In fact, you don't need this acronym at all. It is TCCON jargon and only used three times in the whole manuscript (and you spelled it out each time!).

Response: We made correction "SVI" by "SIV".

- p. 2, l. 27-28: there is no need to provide an outline of the sections and numbers. Scientific papers typically don't have a table of contents. Just drop these two lines.

Response: We improved it. Please see the last paragraph of the introduction section in revised manuscript.

- p. 3, l- 3: Please replace "G-b" with "g-b" throughout the text unless it starts a sentence.

Response: We corrected it.

- p. 3, l. 5: "... Seoul, the capital city of Korea." -> I don't want get into politics here but isn't the country officially named "Republic of Korea"? "South Korea" would probably also be clear, maybe even clearer to the general reader.

Response: Thanks for the comment. The country official name is "Republic of Korea". We corrected the sentence "… Seoul, the capital city of Republic of Korea."

- p. 3, l. 22-23: avoid the line break for "A 547". In fact, I believe the tracker model number is "A547" (w/o space).

Response: Replaced "A 547" by "A547".

- p. 3, l. 23: "... are about 0_ to 315_ and 10_ to 85_ degrees ..." -> (1) "_" and "degrees" are redundant, (2) is the elevation range really only 10 to 85 degrees? Does that mean the tracker cannot point to the horizon or zenith at all?

Response: We removed the redundant of "_" and "degrees" from the text. The tracker can point to the horizon and zenith as well.

- p. 4, l. 3: "oil-free" -> "vacuum pump" is missing. Is the pump running continuously?

Response: The vacuum pump is running continuously.

- p. 5, l. 9: "... voltage ranges of approximately 0 to 219 mV." This information is hardly useful for anyone outside your department. Especially since you claim that "... the detail characteristic of the operation is beyond the scope of this paper."

Response: We omitted "…voltage ranges of approximately 0 to 219 mV…." from the text, since the detail characteristic of the operation is beyond the scope of this paper.

- p. 5, l. 13-14: "the intensity of the incoming light occurred due to changes in thin clouds and aerosols loads or interceptions by any other objects along the line of sight over the measurement site." -> thin clouds is clear but aerosol load should not really change during a 2-minute measurement. And what objects could be passing the line of sight often enough to justify such a system?

Response: We understood that this sentence "…interceptions by any other objects along the line of…." is irrelevant so that we omitted the sentence "…interceptions by any other objects …." in the text.

- p. 6, l. 13: GGG is not developed "by JPL" even though the main developer works there. But there are also other main developers who work at different institutions even outside Caltec/JPL. It would be correct to say that GGG is developed by the TCCON community.

Response: We thank the referee's comments. We corrected it accordingly.

- p. 8, l. 9: "LINFIT" -> "LINEFIT"!
Response: Corrected "LINFIT" by "LINEFIT"

**Figures**
**Response: All figures have been replaced. Data analysis period and quality improved.**

- Fig. 1: (1) picture quality is not very good, (2) country borders and maybe the location of Seoul would be helpful, (3) For the insets: the upper one is clear but what is the lower one? The labels are Korean only.

- Fig. 2: (1) "server" instead of "sever", (2) you used "solar tracker" throughout the text, so you should not use "sun tracker" in the figure, (3) "Photographs of the automated FTS laboratory."

- Fig. 3: how is signal-to-noise defined here?

- Fig. 4: low quality/resolution

- Fig. 5: (1) low quality/resolution, (2) is this the same day for both plots? (3) why does signal drop off to the right with OASIS even though start time is earlier? (3) better plot this over solar zenith angle than over time!

- Fig. 6: very low quality with obvious JPEG compression artifacts. This should be redone in a lossless compression format like PNG or a vector format like PDF!

- Fig. 7: (1) not referenced in the text at all! (2) Should probably belong to Sec. 2.4 which means it should appear before (!) Fig. 6. (3) I don't know why this Figure is even part of the manuscript.

Is this an original figure created by the authors or taken from somewhere else? (4) similar quality problem as with the other figures. The box labels are basically unreadable.

- Fig. 8: (1) low quality/resolution (2) why not just plot XCO2? The variations in column are mostly due to seasonal airmass variation (as can be seen in O2 column). XCO2 would tell you something about carbon-cycle related effects at your site!

- Fig. 9: unlike the other figures, this one has acceptable quality. I would still suggest to plot daily medians instead of means.

**Second referee revision**

[revised manuscript text omitted]

- Fig. 1: (1) picture quality is not very good, (2) country borders and maybe the location of Seoul would be helpful, (3) For the insets: the upper one is clear but what is the lower one? The labels are Korean only.

**2.2 G-b FTS instrument**

Solar spectra are acquired by operating a Bruker IFS 125HR spectrometer (Bruker Optics, Germany) under the framework of TCCON. Currently, our g-b FTS instrument operation is semi-automated for taking the routine measurements under clear sky conditions. It is planned to make an FTS operation mode to be fully automated by this year. The solar tracker (Tracker A547, BrukerOptics, Germany) is mounted inside a dome. The tracking ranges in terms of both azimuthal and elevation angles are about 0 to 315 and 10 to 85 degrees, respectively, while the tracking speed is about 2 degrees per second. The tracking accuracy of ±4 minutes of arc can be achieved by the Camtracker mode. Under clear sky conditions, the dome is opened and set to an automatic-turning mode, so that the mirrors are moved automatically to search for the position where the sunspot is seen by the camera. Then, the solar tracker is activated in such a way that the mirrors are finely and continuously controlled to fix the beam into the spectrometer. Figure 2 displays an overview of the general data acquisition system. This ensures that all spectra were recorded under clear weather conditions. The other important feature that has been made on the FTS spectrometer is the implementation of the interferogram sampling method (Brault, 1996), that takes advantage of modern analog-digital converters (ADCs) to improve the signal-to-noise ratio.

[Figure]

[Figure]

**Figure 11:** Photographs of the automated FTS laboratory. The Bruker Solar Tracker type A547 is mounted in the custom made dome. A servo controlled solar tracker directs the solar beam through a CaF$_2$ window to the FTS (125HR) in the laboratory. The server computer is used for data acquisition. PC1, PC2 and PC3 are used for controlling the spectrometer, solar tracker, dome, camera, pump, GPS satellite time, and humidity sensor.

- Fig. 2: (1) "server" instead of "sever", (2) you used "solar tracker" throughout the text, so you should not use "sun tracker" in the figure, (3) "Photographs of the automated FTS laboratory."

The spectrometer has equipped with two room temperature detectors; an Indium-Gallium-Arsenide (InGaAs) detector, which covers the spectral region from 3,800 to 12,800 cm$^{-1}$, and a Silicon (Si) diode detector (9,000 – 25,000 cm$^{-1}$) used in a dual-acquisition mode with a dichroic optic (Omega Optical, 10,000 cm$^{-1}$ cut-on). A filter (Oriel Instruments 59523; 15,500 cm$^{-1}$ cut-on) prior to the Si diode detector blocks visible light, which would otherwise be aliased into a near-infrared spectral domain. TCCON measurements are routinely recorded at a maximum optical path difference (OPD$_{max}$) of 45 cm leading to a spectral resolution of 0.02 cm$^{-1}$. Two scans, one forward and one backward, are performed and individual interferograms are recorded. A single scan in one measurement takes about 110 s. The pressure inside FTS is kept at 0.1 to 0.2 hPa

with vacuum pump to maintain the stability of the system and to ensure clean and dry conditions.

[Figure]

**Figure 12:** Single spectrum recorded on 4 October 2014 with a resolution of 0.02 cm$^{-1}$. A typical example for the spectrum of $XCO_2$ is shown in the inset.

**Table 6.** Measurement setting for the Anmyeondo g-b FTS spectrometer of the Bruker 125HR model

| Item | Setting |
|---|---|
| Aperture | 0.8 mm |
| Detectors | RT-Si Diode DC, RT-InGaAs DC |
| Beamsplitters | CaF$_2$ |
| Scanner velocity | 10 kHz |
| Low pass filter | 10 kHz |
| High folding limit | 15798.007031 |
| Spectral Resolution | 0.02 cm$^{-1}$ |
| Optical path difference | 45 cm |
| Acquisition mode | Single sided, forward-backward |
| Sample scan | 2 scans |
| Sample scan time | ~110 s |

**2.3 Operational Automatic System for the Intensity of Sunray (OASIS)**

The OASIS system is developed for improving the quality of the spectra recorded by the spectrometer. To ensure the quality of the spectra, this system is beneficial for minimizing the noise that induced in the spectra due to rapid intensity fluctuations of the incoming solar radiation that reaches to the instrument. The main function of the OASIS is to control the aperture diameter of inlet through which the incoming radiation goes to the interferometer. This aperture is placed inside the OASIS system, which is different from the actual aperture that is located inside the interferometer compartment. The aperture size varies in the range of 26 to

32 mm with respect to the photon sensor signals at the OASIS system. Figure 4 depicts the schematic views of the OASIS systems. As can be seen in the figure, the basic components of the OASIS system such as photoelectric sensor, stepping motor, and sunray controller are shown clearly. In fact, the detail characteristic of the operation is beyond the scope of this paper. The fundamental purpose of this system is to optimize the measurement of solar spectra by reducing the effect of the fluctuations (sudden drops) of the intensity of the incoming light occurred due to changes in thin clouds along the line of sight over the measurement site. The maximum threshold value of the solar intensity variation (SIV) is 5 % that is the TCCON standard value (Ohyama et al., 2015). Therefore, we have reduced this value to 2 % in our case by introducing a new home made OASIS system to our g-b FTS since December 2014. This allows us to ensure for having high quality spectra from the instrument. In this work, we have used this quality criterion to screen out the quality of the spectra. Figure 5 illustrates an example, taken on date 4 April 2015, on variations in levels of intensity with and without equipped the OASIS system to the g-b FTS instrument. It is clearly seen that the large amplitude of the solar intensity variation is filtered in the spectra. Note that the solar intensity difference was exhibited as can be seen in the figure, which was due to the measurement time difference.

[Figure]

**Figure 13:** a) Shows the configuration of installed equipment and the path of solar beam and (b) Schematic views of the OASIS system.

- Fig. 4: low quality/resolution

[Figure]

**Figure 14:** Typical example for solar intensity versus time with and without OASIS is given. (Taken on 04 April, 2015)

- Fig. 5: (1) low quality/resolution, (2) is this the same day for both plots? (3) why does signal drop off to the right with OASIS even though start time is earlier? (3) better plot this over solar zenith angle than over time!

**Table 2.** Spectral windows used for the retrievals of the columns of $CO_2$ and $O_2$.

| Gas | Center of spectral window (cm$^{-1}$) | Width (cm$^{-1}$) | Interfering gas |
|---|---|---|---|
| $O_2$ | 7885.0 | 240.0 | $H_2O$, HF, $CO_2$ |
| $CO_2$ | 6220.0 | 80.0 | $H_2O$ ,HDO, $CH_4$ |
| $CO_2$ | 6339.5 | 85.0 | $H_2O$ ,HDO |

[Figure]

**Figure 15:** Modulation efficiency and phase error (rad) of HCl measurements from the g-b FTS are displayed in the period from October 2013 to September, 2014. Resolution: 0.015 cm$^{-1}$, Aperture: 0.8 mm, and Detector: RT-InGaAs DC (from 2013.10 (red) to 2016.09 (black)).

- Fig. 6: very low quality with obvious JPEG compression artifacts. This should be redone in a lossless compression format like PNG or a vector format like PDF!

**2.5 Characterization of FTS-instrumental line shapes**

For the accurate retrieval of total column values of the species of interest, a good alignment of the g-b FTS is essential. The instrument line shape (ILS) is retrieved from the regular HCl cell measurement that is an important indicator of the status of the FTS's alignment (Hase et al., 1999). The analyses of the measurements were performed using a linefit spectrum fitting algorithm (LINEFIT14 software) (Hase et al., 2013). Here, we have carried out experiments to investigate the influences of ILS with and without to the presence of OASIS system, and then we considered HCl cell measurements using sun as source while OASIS system active and tungsten lamp as a source while OASIS inactive. Without OASIS system, we showed the time series of the modulation efficiency and phase error (rad) in the HCl measurement using the source of light from tungsten lamp in the period of October 2013 to September 2016, which is depicted in Fig.

6. Modulation amplitudes for well alignment should be controlled in a limit of 5 % loss at the maximum optical difference (Wunch et al., 2011). In our g-b FTS measurements, it is found that the maximum loss of modulation efficiency at the maximum OPD is about 3 %, which is quite close to the ideal value. The phase errors are less than 0.009. Hase et al. (2013) reported that this level of small disturbances from the ideal value of the modulation efficiency is common to all well-aligned instruments. This result confirmed that the g-b FTS instrument is well aligned and stable during the whole operation period.

In the case OASIS system in active mode, we also confirmed that the ILS was not affected by the variable aperture during the operation of this system. The modulation efficiency and phase error were estimated to be 99.96 % and 0.009 rad, respectively (see Table 3). Sun et al. (2017) reported the detailed characteristics of the ILS with respect to applications of different optical attenuators to FTIR spectrometers within the TCCON and NDACC networks. They used both lamp and sun cell measurements which were conducted after the insertion of five different attenuators in front of and behind the interferometer. In Sun et al. (2017) paper, the ILS result was indicated by considering optical attenuator no .1 which is in good agreement with our findings.

**Table 3.** ILS measurements with and without OASIS system (sources of light are tungsten lamp and solar light).

| Light Source | Tungsten | Solar(Sun) | Solar(Sun) | Range |
|---|---|---|---|---|
| S/N (signal to noise ratio) | 183.2 : 1 | 162.7 : 1 | 167.1 : 1 | _ |
| Center wavenumber | 5687.65 cm$^{-1}$ | 5687.65 cm$^{-1}$ | 5687.65 cm$^{-1}$ | |
|  |  |  |  |  |
| Mod. eff | 99.99 % | 99.98 % | 99.96 % | 99.96 ~99.99 % |
| Phase error (rad) | 0.007 | 0.009 | 0.009 | 0.007 – 0.009 |
| OASIS run | **OFF** | **OFF** | **ON** | _ |
| Parameter | Spectral Resolution: 0.015cm$^{-1}$, Scans: 50, Beamsplitter: CaF$_2$, Aperture: 0.8 mm, Detector: RT-InGaAs DC, Scanner velocity: 10 kHz, High pass filter: open, Low pass filter: 10 kHz, Optical Path Difference (OPD) = 45 cm  Source setting: Emission back parallel input/ NIR | | | |

[Figure]

**Figure 16**: Time series of LSE (top panel) and $X_{air}$ (bottom panel) from the g-b FTS during 2014-2016. Each marker represents a single measurement.

**2.6 Data processing**

Within the TCCON standard retrieval strategy, we have derived the column-averaged dry-air mole fraction $CO_2$ ($XCO_2$) and other atmospheric gases using GFIT algorithm. In this work, the TCCON standard GGG2014 (version 4.8.6) retrieval software was used to obtained abundance of the species from FTS spectra (Wunch et al., 2015). However, there is a slightly different setup of instrumentation in Anmyondo FTS site where all spectra are recorded after the OASIS system equipped, which is described a little bit in section…The $XCO_2$ is the ratio of retrieved $CO_2$ column to retrieved $O_2$ column ,

$$XCO_2 = \frac{CO_{2\ column}}{O_{2\ column}} \times 0.2095 \ ,$$
        (1)

Computing the ratio using Eq. (1) minimizes systematic and correlated errors such as errors in solar zenith angle, surface pressure, and instrumental line shape that existed in the retrieved $CO_2$ and $O_2$ columns (Messerschmidt et al., 2012, Washenfelder et al., 2006). Top panel of Fig.7 depicts the time series of LSE obtained from InGaAs spectra at Anmyondo FTS station in the measurement period of 2014 to 2016. We conducted the laser adjustment or laser replacement on 10 March, 2014, at which large LSE values were shown

(see top panel of Fig. 7).

The $X_{air}$ is a useful indicator of the quality of measurements and the instrument performance. The $X_{air}$ would be unity for an ideal retrieval, however, due to spectroscopic limitations there is a TCCON wide bias and solar zenith angle (SZA) dependence. The retrieval of $X_{air}$ deviating more than 1% from the nominal value of 0.98 would suggest a systematic error. The time series of $X_{air}$ are shown in the bottom panel of Fig. 7. The $X_{air}$ record reveals that the instrument has been stable during the measurement period. It shows that the values of $X_{air}$ are fluctuated between 0.974 and 0.985, and the mean value is 0.982 with a standard deviation of 0.0015 in which the scatter for $X_{air}$ is about 0.15 %. The low variability in time series of $X_{air}$ indicates the stability of the measurements.

**2.6 OCO-2**

Orbiting Carbon Observatory-2 (OCO-2) is NASA's first Earth-orbiting satellite, which was successfully launched on July 2, 2014 into low-Earth orbit. It is devoted to observing atmospheric carbon dioxide ($CO_2$) to get better insight for the carbon cycle. The primary mission is to measure carbon dioxide with high precision and accuracy in order to characterize its sources and sinks at different spatial and temporal scales (Boland et al., 2009; Crisp, 2008, 2015). The instrument measures the near infrared spectra (NIR) of sunlight reflected off the Earth's surface. Using a retrieval algorithm, it provides results of atmospheric abundances of carbon dioxide and related atmospheric parameters at the nadir, sun glint and targets modes. Detailed information about the instrument is available in different papers (Connor et al., 2008; O'Dell et al., 2012). In this work, we used the OCO-2 version 7Br bias corrected data.

[Figure]

**Figure 8**: Time series of $CO_2$ (top panel) and $O_2$ (middle panel) column amounts and surface pressure (bottom right panel) from the g-b FTS are depicted during 2014- 2016. All results are on basis of daily median basis.

[Figure]

**Figure 9.** Time series of XH₂O, XN₂O, XCO, XCH₄, and XCO₂ from top to bottom panels (a-e), respectively in the period between 2014 - 2016. Each marker indicates a single retrieval.

**3 Results and discussion**

**3.1 Time series of g-b FTS columns of $CO_2$ and $O_2$**

The $XCO_2$ along with other retrievals g-b FTS are presented in Fig. 8 (panel e), in the time period of 2014 – 2016. We also incorporated time series of other greenhouse gases (such as $XH_2O$, $XN_2O$, XCO, and $XCH_4$) that are retrieved together with the $XCO_2$, which are depicted in Fig 9.(panel a-d). The temporal distributions of the g-b FTS total column amounts of $CO_2$ and $O_2$ on daily median basis during the period from February 2014 to December 2016 are depicted in the left bottom and right top panels of Fig. 8, respectively. It was shown that the $CO_2$ column amounts varied within $8.40 \times 10^{21}$ to $8.84 \times 10^{21}$ molecules cm$^{-2}$ during the whole observation period, while $O_2$ varied between $4.5 \times 10^{24}$ and $4.7 \times 10^{24}$ molecules cm$^{-2}$, with the corresponding mean of $4.52 \times 10^{24}$ molecules cm$^{-2}$ and a standard deviation of $2.59 \times 10^{22}$ molecules cm$^{-2}$, respectively. The scatter for column $O_2$ is estimated to be 0.57 %,

which is comparable with the variation of atmospheric pressure (see Fig. 8 right top and bottom panels).

[Figure]

**Figure 10.** Time series of XCO₂ retrieval (top left panel) and its retrieval error (bottom left panel) from Anmyeondo FTS and Saga FTS in the period of 2014 – 2016. Top right panel depicts map of TCCON sites which are close to our site.

**3.2 Comparison of Anmyeondo XCO₂ with nearby TCCON site**

We compared our FTS XCO₂ data with similar ground-based high resolution FTS observations at Saga TCCON station (33.26 N, 130.29 E) in Japan, which is the closest TCCON station to our site (see right panel of Fig 10). Among those TCCON sites, Rikubetsu, Tsukuba, and Saga are located in Japan and Hefei is located in China (Wang et al., 2017). To demonstrate the comparison between them, we have shown the daily averaged XCO₂ of two sites during the period of 2014 to 2016 in Fig. 10 left panel. As can be seen, variations of XCO₂ at the Saga site agreed well with Anmyeondo site. The daily averaged XCO₂ revealed the same seasonal cycle as that of our site.

The lowest XCO$_2$ appeared in late summer (August and September), and the highest value was in spring (April).

**3.3 Comparison of XCO$_2$ between the g-b FTS and OCO-2**

In this section, we present a comparison of XCO$_2$ between the g-b FTS and OCO-2 version 7Br data (bias corrected data) over Anmyeondo station during the period between 2014 and 2016. For making a direct comparison of the g-b FTS measurements against OCO-2, we applied the spatial coincidence criteria for the OCO-2 data within 3° latitude/longitude of the FTS station, as well as setting up a time window of 3 hours. Based on the coincidence criteria, we obtained thirteen (13) coincident measurements, which were not sufficient to infer a robust conclusion. But it gives a preliminary result for indicating a level of agreement between them. We showed that the comparison of the time series XCO$_2$ concentrations derived from the g-b FTS and OCO-2 on daily medians basis along with the time series of its retrieval errors from FTS during the measurement period between 2014 and 2016, as depicted in Fig. 10. As can be seen in the plot, the g-b FTS measurement exhibits some gaps occurred due to bad weather conditions, instrument failures, and absences of an instrument operator. In the present analysis, the XCO$_2$ concentrations from FTS were considered only when its retrieval error was below 1.5 ppm (see the bottom panel of Figure 8), which is the sum of all error components such as laser sampling error, zero level offsets, ILS error, smoothing error, atmospheric apriori temperature, atmospheric apriori pressure, surface pressure, and random noise. Recently, Wunch et al. (2016) reported that the comparison of XCO$_2$ derived from the OCO-2 version 7Br data against a co-located ground-based TCCON data that indicates the median differences between the OCO-2 and TCCON data were less than 0.5 ppm, a corresponding RMS differences less than 1.5 ppm. The overall results of our comparisons are comparable with the report made by Wunch et al. (2016). The OCO-2 product of XCO$_2$ was biased (satellite minus g-b FTS) with respect to the g-b FTS, which was slightly higher by 0.179 ppm with a standard deviation of 1.194 ppm. This bias could be attributed to the instrument uncertainty. In addition to that, we also obtained a strong correlation between them, which was quantified as a correlation coefficient of 0.936 (see Table 2).

**Table 4.** Summary of the statistics of XCO$_2$ comparisons between OCO-2 and the g-b FTS from 2014 to 2016 are presented. N –coincident number of data, R - Pearson correlation coefficient, RMSE - Root Mean Squares Error.

| N | Mean Absolute. diff. (ppm) | Mean Relative diff (%) | R | RMSE (ppm) |
|---|---|---|---|---|
| 13 | 0.179±1.194 | 0.0443±0.298 | 0.936 | 1.161 |

[Figure]

**Figure 11:** Left panel: The time series of $XCO_2$ from the g-b FTS (blue triangle) and OCO-2 (red triangle) over the Anmyeondo site from 2014 to 2016. Right panel: The linear regression curve between FTS and OCO-2. All results are given on daily medians basis.

[Figure]

**Figure 12:** Left panel: The time series of $XCO_2$ on monthly mean basis, whereas left panel depicted annual cycle of $XCO_2$.

**3.4 Seasonal cycle of $XCO_2$**

In this section, the main focus of this issue is to deal with the comparison of the seasonal cycle of $XCO_2$ between the g-b FTS and OCO-2 over the Anmyeondo station. In order to understand the role of local influence, we have tried to show the seasonal and annual cycle of $CO_2$ derived from in-situ tower observation. Fig. 12 exhibits the time series of the monthly mean $XCO_2$ and annual cycle for the measurement period of 2014 to 2016 from FTS (blue), OCO-2 (red) and in-situ tower (green solid lines with dot marker). The overall result indicates that both instruments are generally agreed in capturing the seasonal variability of $XCO_2$ at the measurement site. As it is clearly seen from the temporal distribution of FTS $XCO_2$, the maximum and minimum values are observed in spring and late summer seasons, respectively. It was found that its mean values in spring and summer were 402.72 and 396.92 ppm, respectively (see Table 5). This is because the seasonal variation of $XCO_2$ is controlled mainly by the photosynthesis in the terrestrial ecosystem, and this explains the larger $XCO_2$ values in the northern hemisphere in late April (Schneising et al. 2008, and references therein). The minimum value of $XCO_2$ occurs in August, which is most likely due to uptake of carbon into the biosphere in associated with the period of plant growth. Furthermore, both instruments showed high standard deviations during summer, about 3.28 ppm in FTS and 3.77 ppm in OCO-2, suggesting that the variability reflects strong sources and sink signals. However, photosynthesis is not the only driver of the seasonal cycle during the local growing season. The site is also influenced by regional anthropogenic emissions under the prevailing winds.

[revised manuscript text omitted]
, L., Gordon, I., Babikov, Y., Barbe, A., Benner, D. C., Bernath, P., Birk, M., Bizzocchi, L., Boudon, V., Brown, L., Campargue, A., Chance, K., Cohen, E., Coudert, L., Devi, V., Drouin, B., Fayt, A., Flaud, J.-M., Gamache, R., Harrison, J., Hartmann, J.-M., Hill, C., Hodges, J., Jacquemart, D., Jolly, A., Lamouroux, J., Roy, R. L., Li, G., Long, D., Lyulin, O., Mackie, C., Massie, S., Mikhailenko, S., Müller, H., Naumenko, O., Nikitin, A., Orphal, J., Perevalov, V., Perrin, A., Polovtseva, E., Richard, C., Smith, M., Starikova, E., Sung, K., Tashkun, S., Tennyson, J., Toon, G., Tyuterev, V., and Wagner, G.: The HITRAN2012 molecular spectroscopic database, J. Quant. Spectrosc. Ra., 130, 4–50, doi:10.1016/j.jqsrt.2013.07.002, 2013.

Schneising O., Buchwitz M., Burrows J. P., Bovensmann, H., Reuter, M., Notholt, J., Macatangay, R., and Warneke, T.: Three years of greenhouse gas column-averaged dry air mole fractions retrieved from satellite-Part 1: Carbon dioxide, Atmos. Chem. Phys., 8, 3827-3853, 2008.

Sun, Y., Palm, M., Weinzierl, C., Peteri, C., Notholt, J., Wang, Y., and Liu, C.: Technical note: Sensitivity of instrumental line shape monitoring for the ground-based high-resolution FTIR spectrometer with respect to different optical attenuators, Atmos. Meas. Tech. Discuss., doi:10.5194/amt-10-989- 2017.

Wang, W., Tian, Y., Liu, C., Sun, Y., Liu, W., Xie, P., Liu, J., Xu, J., Morino, I., Velazco, V., Griffith, D., Notholt, J., and Warneke, T.: Investigating the performance of a greenhouse gas observatory

in Hefei, China, Atmos. Meas. Tech., 10, 2627–2643, 2017.

Washenfelder R. A., Toon G. C., Blavier J-F., Yang Z., Allen N. T., Wennberg P. O., Vay S. A., Matross, D. M., and Daube B. C.: Carbon dioxide column abundances at the Wisconsin Tall Tower site, Journal of Geophysical Research, 2006, 111, doi:10.1029/2006JD00715, 2000. Warneke, T., Yang, Z., Olsen, S., Korner, S., Notholt J., Toon, G. C., Velazco, V., Schultz, A., and Schrems, O.: Seasonal and latitudinal variations of column averaged volume-mixing ratios of atmospheric $CO_2$, Geophysical Research Letters, 32(3), 2-5, doi:10.1029/2004GL021597, 2005.

Wunch, D., Toon G. C., Wennberg, P. O., Wofsy, S. C., Stephens, B., Fisher, M. L., Uchino O., Abshire, J. B., Bernath, P. F., Biraud, S. C., Blavier, J.-F. L., Boone, C. D., Bowman, K. P., Browell, E. V., Campos, T., Connor, B. J., Daube, B. C., Deutscher, N. M., Diao M., Elkins, J. W., Gerbig, C., Gottlieb, E., Griffith, D. W. T., Hurst, D. F., Jiménez, R., Keppel-Aleks, G., Kort, E. A., Macatangay, R., Machida, T., Matsueda, H., Moore, F. L., Morino, I., Park, S., Robinson, J., Roehl, C. M., Sawa, Y., Sherlock, V., Sweeney, C., Tanaka, T., and Zondlo, M. A.: Calibration of the Total Carbon Column Observing Network using aircraft profile data, Atmospheric Measurement Techniques, 3(5), 1351-1362, doi:10.5194/amt-3-1351-2010.

Wunch, D., Toon, G. C., Blavier, J.-F. L., Washenfelder, R. A., Notholt, J., Connor, B. J., Griffith, D. W. T., Sherlock, V., and Wennberg, P. O.: The Total Carbon Column Observing Network, Philos. T. R. Soc. A, 369, 2087–2112, doi:10.1098/rsta.2010.0240, 2011.

Wunch, D., Toon, G. C., Sherlock, V.,  Deutscher, N. M., Liu X., Feist, D. G., and Wennberg, P. O.: The Total Carbon Column Observing Network's GGG2014 Data Version. doi:10.14291/tccon.ggg2014.documentation.R0/1221662, 2015.

Wunch, D., Wennberg, P.O.,, Osterman, G., Fisher, B., Naylor, B., Roehl, C. M., O'Dell, C., Mandrake, L.,Viatte, C., Griffith, D. W. T., Deutscher, N. M., Velazco, V. A., Notholt, J.,Warneke, T.,Petri, C., Maziere, M. De, Sha, M. K., Sussmann,R., Rettinger, M., Pollard, D., Robinson, J.,Morino, I., Uchino O., Hase, F., Blumenstock, T., Kiel, M., Feist, D. G., Arnold S.G., Strong, K., Mendonca, J., Kivi, R.,Heikkinen, P., Iraci, L., Podolske, J., Hillyard, P. W., Kawakami, S., Dubey, M. K., Parker, H. A., Sepulveda, E., Rodriguez, O. E. G., Te, Y., Jeseck, P., Gunson, M. R., Crisp, D., and Eldering A., Comparisons of the Orbiting Carbon Observatory-2 (OCO-2) $XCO_2$ measurements with TCCON, Atmos. Meas. Tech. Discuss., doi:10.5194/amt-2016-227, 2016

**Co-authors and TCCON  comments.**

- You can also add the comparison with the KORUS-AQ in situ profile to strengthen the paper.

- Then write  a separate paper focussed on OASIS, using these measurements with and without OASIS.

Response: I will follow the TCCON measurement guidelines. I will re-work my existing paper without OASIS content.( I will follow your suggestion). , I will write a separate paper for the OASIS section through further study. The KORUS-AQ in situ profile add in the paper.

**Final comment revision. (Final version.)**

**Characteristics of the Greenhouse Gas Concentration Derived from the Ground-based FTS Spectra at Anmyeondo, Korea**

Young-Suk Oh[1,2*], Samuel Takele Kenea[1], Tae-Young Goo[1], Kyu-Sun Chung[2], David W. T. Griffith[3], Paul. Wennberg[4], Voltaire A. Velazco[3], Jae-Sang Rhee[1], Mi-Lim Ou[5], and Young-Hwa Byun[1]

1. Climate Research Division, National Institute of Meteorological Sciences (NIMS), Jeju–do, Republic of Korea
2. Department of Electrical Eng. & Centre for Edge Plasma Science, Hanyang University, Seoul, Republic of Korea
3. School of Chemistry, University of Wollongong, Wollongong, Australia
4. California Institute of Technology, California, USA
5. Climate Change Monitoring Division, Korea Meteorological Administration, Seoul, Republic of Korea

*Correspondence to: Young–Suk Oh (ysoh306@gmail.com )

**Abstract.**

Since the late 1990s, the meteorological observatory established in Anmyeondo (36.5382° N, 126.3311° E, and 30 m above mean sea level), has been monitoring several greenhouse gases such as $CO_2$, $CH_4$, $N_2O$, CFCs, and $SF_6$, as part of the Global Atmosphere Watch (GAW) Program. A high resolution ground-based (g-b) Fourier Transform Spectrometer (FTS, IFS-125HR model) was installed at such observation site in 2013, and has been fully operated within the frame work of the Total Carbon Column Observing Network (TCCON) since August, 2014. The solar spectra recorded by the g-b FTS are covered in the range between 3,800 and 16,000 $cm^{-1}$ at the spectral resolution of 0.02 $cm^{-1}$ during the measurement period between 2014 and 2016. In this work, the GGG2014 version of the TCCON standard retrieval algorithm was used to retrieve $XCO_2$ concentrations from the FTS spectra. Two spectral bands (at 6220.0 and 6339.5 $cm^{-1}$ centre wavenumbers) were used to derive the $XCO_2$ concentration within the spectral residual of +0.01 %. All sources of errors were thoroughly analyzed. In this paper, we introduced aircraft observation campaigns over Anmyeondo station were carried out during the period between 2012 and 2016. A comparison of the $XCO_2$ concentration in g-b FTS and OCO-2 (Orbiting Carbon Observatory) satellite observations was presented only for the measurement period between February 2014 and December 2016. The 13 coincident observations were selected on a daily median basis. It was obtained that OCO-2 exhibited slightly higher 
[revised manuscript text omitted]

tungsten lamp in the period of October 2013 to September 2017, which is depicted in Figure 4. Modulation amplitudes for well alignment should be controlled in a limit of 5 % loss at the maximum optical difference (Wunch et al., 2011). In our g-b FTS measurements, it is found that the maximum loss of modulation efficiency is less than 1 %, which is quite close to the ideal value. The phase errors are less than 0.0001. Hase et al. (2013) reported that this level of small disturbances from the ideal value of the modulation efficiency is common to all well-aligned instruments. This result confirmed that the g-b FTS instrument is well aligned and stable during the whole operation period.

[Figure]

**Figure 4.** Phase error (rad) (left panel) and Modulation efficiency (right panel) of HCl measurements from the g-b FTS are displayed in the period from October 2013 to September 2017. Resolution = 0.015 cm$^{-1}$, Aperture = 0.8 mm.

We also confirmed that the ILS was not affected by the variable aperture during the operation of this system. The modulation efficiency and phase error were estimated to be 99.98 % and 0.0001 rad. Sun et al. (2017) reported the detailed characteristics of the ILS with respect to applications of different optical attenuators to FTIR spectrometers within the TCCON and NDACC networks. They used both lamp and sun cell measurements which were conducted after the insertion of five different attenuators in front of and behind the interferometer. In Sun et al.

(2017), the ILS result was indicated by considering optical attenuator number 1 which is in good agreement with our findings.

**Table 2.** Spectral windows used for the retrievals of the columns of $CO_2$ and $O_2$.

| Gas | Center of spectral window (cm$^{-1}$) | Width (cm$^{-1}$) | Interfering gas |
|-----|------|------|------|
| $O_2$ | 7885.0 | 240.0 | $H_2O$, HF, $CO_2$ |
| $CO_2$ | 6220.0 | 80.0 | $H_2O$ ,HDO, $CH_4$ |
| $CO_2$ | 6339.5 | 85.0 | $H_2O$ ,HDO |

**2.4 Data processing**

Within the TCCON standard retrieval strategy, we have derived the column-averaged dry-air mole fraction $CO_2$ ($XCO_2$) and other atmospheric gases ($O_2$, CO, $CH_4$, $N_2O$, and $H_2O$) using GFIT algorithm. The spectral windows used for the retrieval of $CO_2$ and $O_2$ are given in Table 2. The TCCON standard GGG2014 (version 4.8.6) retrieval software was used to obtain the abundance of the species from FTS spectra (Wunch et al., 2015).  The $XCO_2$ is the ratio of retrieved $CO_2$ column to retrieved $O_2$ column,

$$XCO_2 = \frac{CO_{2 \text{ column}}}{O_{2 \text{ column}}} \times 0.2095 , \qquad (1)$$

Computing the ratio using Eq. (1) minimizes systematic and correlated errors such as errors in solar zenith angle, surface pressure, and instrumental line shape that existed in the retrieved $CO_2$ and $O_2$ columns (Washenfelder et al., 2006, Messerschmidt et al., 2012). Top panel of Fig.6 depicts the time series of laser sampling error (LSE) obtained from InGaAs spectra at the Anmyeondo FTS station in the measurement period of February 2014 to December 2016. LSE is small and centered around zero in an ideal case. Slightly large LSE values were shown on 10 March, 2014 (see top panel of Fig. 7). On this date, we conducted the laser adjustment in FTS.

The $X_{air}$ is a useful indicator of the quality of measurements and the instrument performance. The $X_{air}$ would be unity for an ideal retrieval, however, due to spectroscopic limitations there is a TCCON wide bias and solar zenith angle (SZA) dependence. The retrieval of $X_{air}$ deviating more than 1% from the nominal value of 0.98 would suggest a systematic error. The time series of $X_{air}$ is shown in the bottom panel of Figure 5. The $X_{air}$ record reveals that the instrument has been stable during the measurement period. It

shows that the values of $X_{air}$ are fluctuated between 0.974 and 0.985, and the mean value is 0.982 with a standard deviation of 0.0015 in which the scatter for $X_{air}$ is about 0.15 %. The low variability in time series of $X_{air}$ indicates the stability of the measurements.

[Figure]

**Figure 5.** Time series of LSE (top panel) and $X_{air}$ (bottom panel) from the g-b FTS during 2014- 2016 is shown. Each marker represents a single measurement.

**2.5 Aircraft observation campaigns over Anmyeondo station**

In this section, we have discussed a preliminary comparison results made between aircraft observations and g-b FTS over the Anmyeondo station. The aircraft campaign conducted over Anmyeondo station was monitored by National Institute of Meteorological Sciences (NIMS). The aircraft was equipped with a Wavelength Scanned Cavity Ring Down Spectrometer (CRDS; Picarro, G2401-mc) providing mixing ratio data recorded at 0.3 Hz intervals. The position of the aircraft was monitored by GPS, and information on the outside temperature, static pressure, and ground speed was provided by the aircraft's instruments. Data observed during ascent and descent of the aircraft are considered as vertical profiles of $CO_2$ and $CH_4$ over the measurement station. The temperature and pressure of the gas sample have to be tightly controlled at 45 ℃ and 140 Torr in the CRDS, which leads to highly stable spectroscopic features (Chen et al., 2010). Any deviations from these values cause a reduction of the instrument's precision. Data recorded beyond these range of variations in cavity pressure and temperature were discarded in this analysis. Variance of the cavity pressure and temperature in flight result in variance in the $CO_2$ and $CH_4$ mixing ratios. The Picarro CRDS instrument has been regularly calibrated with respect to the standard gases within the error range recommend by World Meteorological Organization ($CO_2$ is 380.23 ± 0.1 ppm, $CH_4$ is 1.825 ± 0.001 ppm)

Several aircraft observation campaigns over Anmyeondo station were carried out during the period between 2012 and 2016. However, a few numbers of aircraft data matched with the remote sensing instruments were available during this observation period. The total number of the aircraft measurements that matched with g-b FTS was only three and all those coincident observations were laid within a period of 2015. The g-b FTS retrieval of $XCO_2$ and $XCH_4$ were compared with aircraft measurements. Here, FTS data were averaged over a time window of ± 30 minutes with respect to the aircraft measurement time. In addition, the averaging kernel of the FTS was applied into the aircraft data. The g-b FTS data were corrected for an airmass-dependent artefact for $XCO_2$ and $XCH_4$, as well as calibrated with respect to TCCON common scaling factors. This scale factor was derived empirically using aircraft profiles over many TCCON sites in order to place the TCCON data on the WMO standard reference scales (Wunch et al. 2010) for both $XCO_2$ and $XCH_4$. This comparison study will be useful for ensuring that the TCCON common scale factors can be applied to our g-b FTS data. The statistical results for $XCO_2$ and $XCH_4$ comparisons between aircraft and g-b FTS are summarized in Table 3. The mean absolute difference between FTS and aircraft were found to be -0.798 ± 1.734 ppm, the corresponding mean relative differences of -0.196 ± 0.427 % for $XCO_2$, while the mean absolute difference of $XCH_4$ is -0.0079 ± 0.012 ppm, with a corresponding mean relative difference of -0.426 ± 0.632 %. These differences appeared on both species were consistent with the combined total errors of instruments. Wunch et al. (2010) reported that the uncertainties ($2\sigma$) of the TCCON common scale are approximately 0.2 % for $XCO_2$ and 0.4 % for $XCH_4$. It is determined that our g-b FTS uncertainty was found to be within this range of uncertainties and can be calibrated against WMO standard scale. Here, we also include some results from the aircraft campaign conducted in 2016, which was operated by KORUS-AQ (Korea-U.S.-Air Quality) joint program aiming at advancing the ability to monitor air pollution from space. Figure 6 illustrates the results of $XCO_2$ and $XCH_4$ comparisons between the aircraft observation and TCCON sites data. Light blue diamond marks show for Anmyeondo station. Our results laid within the indicated linear regression curves as with other TCCON sites.

**Table 3.** The statistical results for $XCO_2$ and $XCH_4$ comparisons between aircraft and g-b FTS are summarized

| Instruments (Aircraft vs. g-b FTS) | No. of coincident measurement | Absolute difference (ppm) | Relative diff. (%) |
|---|---|---|---|

| | | | |
|---|---|---|---|
| XCO$_2$ | 3 | -0.798 ± 1.734 | -0.196 ± 0.427 |
| XCH$_4$ | 3 | -0.0079 ± 0.012 | -0.426 ± 0.632 |

[Figure]

[Figure]

**Figure 6.** The comparisons of XCO$_2$ and XCH$_4$ between the aircraft observation and TCCON sites data are shown. The left side is XCO$_2$ and the right side is XCH$_4$ (light blue depicts for Anmyeondo station) .

**2.6 OCO-2**

[revised manuscript text omitted]

**3.2 Correlations between $XCO_2$ and XCO**

CO is co-emitted with $CO_2$ from combustion sources, leading to a significant positive correlation between them when combustion is a significant source of observed $CO_2$. The midday peaks for each gas reflect the influence of anthropogenic emissions. To examine this effect, we have determined the correlations between $\Delta XCO$ and $\Delta XCO_2$ at our site. In order to compute the correlations, first we have selected hourly averaged data for both XCO and $XCO_2$ that were recorded between 06:00 and 07:00 UTC (i.e 15:00 and 16:00 LST, local standard time), excluding summer data, and then calculated the anomalies by subtracting the hourly averaged data from the mean of the selected data during the measurement period of February 2014 to December 2016. Figure 9 depicts the relationship between hourly $CO_2$ and CO means of anomalies at Anmyeondo during the whole measurement period, excluding summer data. $CO_2$ and CO had a correlation of 0.50, and this suggests that there is an influence of combustion emissions on $CO_2$. However, in a summer season, a negative relationship between them was identified at this site, with the small magnitude of correlation -0.22, and a correlation slope of -0.84. In Ohyama et al., (2015) paper, they derived the correlation coefficients and slopes of $\Delta XCO/\Delta XCO_2$ and $\Delta XCH_4/\Delta XCO$ in order to understand the short term variations of $XCO_2$, XCO, and $XCH_4$ in summer seasons during July 2011 and December 2014 at Saga, Japan. The trajectories for the summer season were classified into three types, depending on the origin of the air masses. The trajectories for types I, II, and III relate to transport of air masses from the Asian continent (China), Southeast Asia, and the Pacific Ocean, respectively. The negative slope of the $\Delta XCO/\Delta XCO_2$ ratio for the type I (slope was -3.15 ppb ppm$^{-1}$) gentler than for the type II (slope was -14.3 ppb ppm$^{-1}$), which was due to the transport of the air masses that experience the strong biospheric uptake of $CO_2$ over the Asia. This argument could support for our analysis at Anmyeondo station. The slope that we obtained in our station is close to the slope reported in type III case in Ohyama et al., (2015) paper.

[Figure]

**Figure 9.** Correlation between $XCO_2$ versus XCO anomalies at Anmyeondo FTS station between February 2014 and December 2016, excluding summer data, is depicted.

In Wang et al (2010), the diurnal cycles of $CO_2$ signal was dominated by the biospheric activity from May to September, with a maximum drawdown of 39 ppmv in daily $CO_2$ in the summer at rural station near Beijing. Biospheric activity, however, has little impact on CO except for the CO source from in situ oxidation of biogenic hydrocarbons. They obtained that the correlation between $CO_2$ and CO in summer was insignificant. The correlation slope gives the emission ratio of CO to $CO_2$, which fluctuates with the sources of $CO_2$, depending on different combustion types and biospheric activity. In our case, the correlation slope of CO to $CO_2$ was found to be 2.27 ppb ppm$^{-1}$ during the whole measurement period excluding summer, which is smaller than the correlation slope reported in Hefei FTS station where it was estimated to be 5.66 ppb ppm$^{-1}$ (Wang et al., 2017 and references therein), which are primarily attributed to the smaller emission in CO.

[revised manuscript text omitted]